
# Conjectures on hidden Onsager algebra symmetries
# in interacting quantum lattice models

**Yuan Miao**

Institute for Theoretical Physics, University of Amsterdam,
Postbus 94485, 1090 GL Amsterdam, The Netherlands

⋆ y.miao@uva.nl

## Abstract

We conjecture the existence of hidden Onsager algebra symmetries in two interacting quantum integrable lattice models, i.e. spin-1/2 XXZ model and spin-1 Zamolodchikov–Fateev model at *arbitrary* root of unity values of the anisotropy. The conjectures relate the Onsager generators to the conserved charges obtained from semi-cyclic transfer matrices. The conjectures are motivated by two examples which are spin-1/2 XX model and spin-1 $U(1)$-invariant clock model. A novel construction of the semi-cyclic transfer matrices of spin-1 Zamolodchikov–Fateev model at arbitrary root of unity values of the anisotropy is carried out via the transfer matrix fusion procedure.

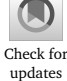

# 1  Introduction

The Onsager algebra was used for the first time to solve two-dimensional Ising model with zero magnetic field by Lars Onsager in his seminal paper in 1944 [1], which is considered to be the herald of exactly solvable models in statistical mechanics.

Later the Onsager algebra has been used to study two-dimensional chiral Potts model, a generalisation of Ising model and its quantum counterpart, $\mathbb{Z}_N$-symmetric spin chain [2–6]. Those models studied via the Onsager algebra are integrable and possess Kramers–Wannier duality [7]. Later, Dolan and Grady used the self-duality of these models to construct infinitely-many conserved charges without invoking their integrability [8]. The equivalence to the Onsager algebra was later discovered by Perk [9]. This approach has led to deeper understanding of the algebraic structure of the Onsager algebra and its relation to self-duality and integrability [10–13]. A thorough and comprehensive summary of the mathematical structures of the Onsager algebra is provided in [14]. Furthermore, an isomorphism between the Onsager algebra and a non-standard classical Yang-Baxter algebra is obtained in [15]. Recently there have been several influential results using the Onsager algebra to study the spectra of quantum lattice models [16], the out-of-equilibrium dynamics of quantum states [17], and the construction of quantum many-body scars [18], reigniting the interest on the applications of the Onsager algebra in theoretical physics.

Meanwhile, the spectra of quantum integrable models at root of unity values of the anisotropy have been investigated using Bethe ansatz techniques. The definition of the parametrisation of root of unity value of the anisotropy is given in (16). Specifically, spin-1/2 XXZ model [19–22] at root of unity, a prototypical quantum integrable lattice model, has drawn lots of attention. Due to the underlying quantum group structure at root of unity [23], the spectra have exponentially many degeneracies. This phenomenon has been studied in [24–30]. More recently, the author and collaborators have constructed the Baxter's Q operator for XXZ model at root of unity and studied the spectra in terms of descendant towers [30], which

elucidated the origin of the exponentially many degeneracies due to the existence of eigenstates associated with exact (Fabricius–McCoy) strings. In particular, XX model, a special case of XXZ model at root of unity, possesses the Onsager algebra symmetry [16] additionally, cf. Sec. 4.1, and its spectrum has similar descendant tower structure as the spectra at other roots of unity [30]. It is thus natural to consider the question whether XXZ models at arbitrary roots of unity value of the anisotropy would possess similar Onsager algebra symmetries. The difficulty of solving this problem is that the Onsager generators in XX model are expressed in terms of local operators, while the generators at other roots of unity are not if they were to exist, which are discussed in Secs. 3 and 4.2. Moreover, it has been shown that the spectra of higher spin generalisations of XXZ model at root of unity, dubbed $U(1)$-invariant clock models, have similar exponentially many degeneracies in terms of descendant tower structure, demonstrated through the Onsager algebra symmetry of the $U(1)$-invariant clock models in [16]. This provides further motivation to the current work. This article is set to compose two conjectures about the explicit form of the Onsager generators of XXZ model and its spin-1 generalisation, i.e. Zamolodchikov–Fateev (ZF) model, at arbitrary root of unity values of the anisotropy and their relations to the conserved charges derived from semi-cyclic transfer matrices.

The structure of the article is as follows. First, we introduce the basic properties of Onsager algebra in Sec. 2. We show an isomorphism between two (Onsager) algebras in the literature. Second, we focus on the spin-1/2 case, i.e. XXZ model at root of unity. We construct the semi-cyclic transfer matrix and the conserved charges thereof in Sec. 3. Motivated by the example of XX model, the conjectures of hidden Onsager algebra symmetries in XXZ model at arbitrary root of unity values of the anisotropy are given in Sec. 4. For the spin-1 case, we construct the semi-cyclic transfer matrices for spin-1 ZF models at arbitrary root of unity values of the anisotropy via transfer matrix fusion procedure in Sec. 5, which has not been reported before. Using another example of the spin-1 $U(1)$-invariant clock model, we formulate similar conjectures of hidden Onsager algebra symmetries in spin-1 ZF models at arbitrary roots of unity in Sec. 6. We conclude the article with Sec. 7.

## 2 Onsager algebra

We use the notations in Onsager's original paper [1] to define the standard presentation of the Onsager algebra $O$. Consider the following infinite-dimensional Lie algebra $O$ with basis $\{\mathbf{A}_m, \mathbf{G}_n | m, n \in \mathbb{Z}\}$. The *canonical* generators satisfy the following relations,

$$[\mathbf{A}_m, \mathbf{A}_n] = 4\mathbf{G}_{m-n}, \quad [\mathbf{G}_m, \mathbf{A}_n] = 2(\mathbf{A}_{n+m} - \mathbf{A}_{n-m}), \quad [\mathbf{G}_m, \mathbf{G}_n] = 0. \tag{1}$$

From the definition, we easily find out that

$$\mathbf{G}_{-m} = -\mathbf{G}_m, \quad \forall m \in \mathbb{Z}. \tag{2}$$

Next, we define another presentation of the Onsager algebra $O'$ with basis $\{\mathbf{A}_m^r | r \in \{0, +, -\}, m \in \mathbb{Z}\}$ [16]. These generators satisfy

$$\begin{aligned} \mathbf{A}_{-m}^0 = \mathbf{A}_m^0, \quad \mathbf{A}_{-m}^\pm = -\mathbf{A}_m^\pm, \\ [\mathbf{A}_m^r, \mathbf{A}_n^r] = 0, \quad m, n \in \mathbb{Z}, r \in \{0, +, -\}, \end{aligned} \tag{3}$$

$$[\mathbf{A}_m^r, \mathbf{A}_n^r] = 0, \ r \in \{0, +, -\}; \quad [\mathbf{A}_m^-, \mathbf{A}_n^+] = \mathbf{A}_{n+m}^0 - \mathbf{A}_{n-m}^0, \tag{4}$$

$$[\mathbf{A}_m^-, \mathbf{A}_n^0] = 2(\mathbf{A}_{n+m}^- - \mathbf{A}_{n-m}^-), \quad [\mathbf{A}_m^+, \mathbf{A}_n^0] = 2(\mathbf{A}_{n-m}^+ - \mathbf{A}_{n+m}^+). \tag{5}$$

From (3), we observe that $\mathbf{A}_0^{\pm} = 0$. We illustrate in Appendix A that presentations $O$ and $O'$ are isomorphic to each other, i.e. they are both presentations of Onsager algebra. Since $O$ is isomorphic to $O'$, we refer to the generators of presentation $O'$ as Onsager generators of $O'$.

As proven by Perk [9] and Davies [10], the Onsager algebra is equivalent to the *Dolan–Grady (DG) relations*, which impose requirements only on the first two generators of $O$, i.e.

$$\Big[\mathbf{A}_0, \big[\mathbf{A}_0, [\mathbf{A}_0, \mathbf{A}_1]\big]\Big] = 16[\mathbf{A}_0, \mathbf{A}_1], \quad \Big[\mathbf{A}_1, \big[\mathbf{A}_1, [\mathbf{A}_1, \mathbf{A}_0]\big]\Big] = 16[\mathbf{A}_1, \mathbf{A}_0]. \tag{6}$$

Due to the isomorphism, the DG relations also impose certain relation between $\mathbf{A}_0^r$ and $\mathbf{A}_1^r$ ($r \in \{0, +, -\}$), which can be obtained using (A.1). We will use the DG relations as the defining property of the existence of the Onsager algebra. Namely, once finding two operators that satisfy the DG relations in certain physical systems, we can construct a family of operators fulfilling the definition (1), which can be considered as a representation of the Onsager algebra $O$. For example, in the case of one-dimensional transverse field Ising model, i.e. the quantum counterpart of two-dimensional Ising model considered by Onsager, we have

$$\mathbf{A}_0 \to \sum_{j=1}^{N} \sigma_j^z, \quad \mathbf{A}_1 \to \sum_{j=1}^{N} \sigma_j^x \sigma_{j+1}^x, \tag{7}$$

satisfying the DG relations (6), thus being a representation of the quotient of Onsager algebra $O$ (1) [10].

**Self-duality**  The DG relations (6) also imply the Kramers–Wannier self-duality [7], which has been used to obtain the value of the phase transition point for models consisting of the Onsager generators. Suppose that the operators $\mathbf{A}_0$ and $\mathbf{A}_1$ can be expressed in local terms, i.e.

$$\mathbf{A}_0 = \sum_{j=1}^{N} \mathbf{a}_{0,j}, \quad \mathbf{A}_1 = \sum_{j=1}^{N} \mathbf{a}_{1,j}. \tag{8}$$

The Kramers–Wannier self-duality implies that the mapping $\mathbf{a}_{0,j} \to \mathbf{a}_{1,j}$ (and conversely $\mathbf{a}_{1,j} \to \mathbf{a}_{0,j}$) leaves the algebraic structure intact, which is obvious from the DG relations (6). Let us consider a Hamiltonian $\mathbf{H}$ that can be expressed in terms of $\mathbf{A}_0$ and $\mathbf{A}_1$,

$$\mathbf{H} = \mathbf{A}_0 + \lambda \mathbf{A}_1, \quad \lambda \in \mathbb{R}, \tag{9}$$

such as one-dimensional transverse field Ising model and chiral Potts model. Using Kramers–Wannier self-duality, one could detect a phase transition at $\lambda = 1$ without solving the entire system [7].

**$U(1)$-invariant Hamiltonian**  In the remaining part of the article, we will focus on a specific type of Hamiltonians that commute with both $\mathbf{A}_0$ and $\mathbf{A}_1$ of the Onsager generators of $O$, i.e.

$$[\mathbf{H}, \mathbf{A}_0] = [\mathbf{H}, \mathbf{A}_1] = 0. \tag{10}$$

These models are referred as $U(1)$-invariant [16], because both operators $\mathbf{A}_0$ and $\mathbf{A}_1$ are considered as $U(1)$ charges of the Hamiltonian. However, $\mathbf{A}_0$ and $\mathbf{A}_1$ do not commute with each other, cf. (1).

From (1), it is easy to observe that $\mathbf{H}$ commutes with all generators in Onsager algebra,

$$\big[\mathbf{H}, \mathbf{A}_m^r\big] = 0, \quad r \in \{0, +, -\}, m \in \mathbb{Z}. \tag{11}$$

Two examples of $U(1)$-invariant Hamiltonians are spin-1/2 XX model and spin-1 ZF model with anisotropy parameter $\eta = \frac{i\pi}{3}$ or $\frac{2i\pi}{3}$. These examples are $U(1)$-invariant clock models

defined in [16]. These two cases will be examined in details in Secs. 4.1 and 6.1, respectively. As explained in latter sections, we conjecture that spin-1/2 XXZ model and spin-1 ZF model at arbitrary root of unity values of the anisotropy belong to the class of $U(1)$-invariant Hamiltonians that possess hidden Onsager algebra symmetries.

A further discussion on the relation between the presentation $O'$ and $\mathfrak{sl}_2$ loop algebra is given in Appendix B.

# 3 Spin-$1/2$ case: semi-cyclic transfer matrices in XXZ model

We consider the $N$-site quasi-periodic (periodic with twist $\phi$) spin-1/2 XXZ Hamiltonian which can be expressed as

$$\mathbf{H}(\phi) = \sum_{j=1}^{N} \left( \frac{1}{2} \left( \sigma_j^+ \sigma_{j+1}^- + \sigma_j^- \sigma_{j+1}^+ \right) + \frac{\Delta}{4} \left( \sigma_j^z \sigma_{j+1}^z - 1 \right) \right). \tag{12}$$

Here $\sigma_j^\pm := (\sigma_j^x \pm i\sigma_j^y)/2$, $\sigma_j^\alpha = \mathbb{1}^{\otimes(j-1)} \otimes \sigma^\alpha \otimes \mathbb{1}^{\otimes(N-j)}$ denotes the $\alpha$th Pauli matrix acting at site $j$ and $\sigma_{N+j}^\pm = e^{\pm i\phi} \sigma_j^\pm$ with $1 \le j < N$. The Hamiltonian is hermitian provided the twist $\phi$ and anisotropy parameter $\Delta$ are real. The latter can be parametrised as

$$\Delta = \frac{q + q^{-1}}{2} = \cosh\eta, \qquad q = e^\eta. \tag{13}$$

The spin-1/2 XXZ Hamiltonian is integrable, allowing us to construct transfer matrices that commute with the Hamiltonian. It can be considered as the Hamiltonian limit of the 6-vertex model [31]. We use transfer matrices as the generating functions for infinitely many conserved charges for the Hamiltonian. Moreover, transfer matrices can be written in terms of Lax operator with auxiliary space labelled by $a$,

$$\mathbf{L}_{aj}(u) = \sinh u \left( \frac{\mathbf{K}_a + \mathbf{K}_a^{-1}}{2} \right) \otimes \mathbb{1}_j + \cosh u \left( \frac{\mathbf{K}_a - \mathbf{K}_a^{-1}}{2} \right) \otimes \sigma_j^z$$
$$+ \sinh\eta \left( \mathbf{S}_a^+ \otimes \sigma_j^- + \mathbf{S}_a^- \otimes \sigma_j^+ \right). \tag{14}$$

The operators in the auxiliary space $a$ satisfy $\mathcal{U}_q(\mathfrak{sl}_2)$ algebra [32],

$$\mathbf{K}_a^2 \mathbf{S}_a^\pm \mathbf{K}_a^{-2} = q^{\pm 2} \mathbf{S}_a^\pm, \quad \left[ \mathbf{S}_a^+, \mathbf{S}_a^- \right] = \frac{\mathbf{K}_a^2 - \mathbf{K}_a^{-2}}{q - q^{-1}}. \tag{15}$$

In this paper, we only consider the root of unity case, i.e.

$$\eta = i\pi \frac{\ell_1}{\ell_2}, \quad q = \exp\left( i\pi \frac{\ell_1}{\ell_2} \right), \quad \varepsilon = q^{\ell_2} = \pm 1, \tag{16}$$

where $\ell_1$ and $\ell_2$ are coprimes. It is obvious that parameter $q$ is a root of unity with $q^{2\ell_2} = 1$. In this case, we use the $\ell_2$-dimensional semi-cyclic representation for the auxiliary space, explicitly given in Appendix C.3. The Lax operator is therefore denoted as $\mathbf{L}^{\mathrm{sc}}$. The semi-cyclic Lax operator $\mathbf{L}_{aj}^{\mathrm{sc}}(u, s, \beta)$ satisfies the following "RLL" relation,

$$\mathbf{R}_{am}^{\mathrm{sc}}(u - v, s, \varepsilon\beta) \mathbf{L}_{an}^{\mathrm{sc}}(u, s, \varepsilon^2\beta) \mathbf{L}_{mn}(v) = \mathbf{L}_{mn}(v) \mathbf{L}_{an}^{\mathrm{sc}}(u, s, \varepsilon\beta) \mathbf{R}_{am}^{\mathrm{sc}}(u - v, s, \varepsilon^2\beta), \tag{17}$$

with $\beta \in \mathbb{C}$. The "RLL" relation is first proven in Section 9 of [30], cf. Eq. (9.14) of [30]. Hilbert spaces $m$ and $n$ both correspond to $\mathbb{C}^2$, and R matrix $\mathbf{R}_{am}^{\mathrm{sc}}(u) = \mathbf{L}_{am}^{\mathrm{sc}}(u + \eta/2)$. When

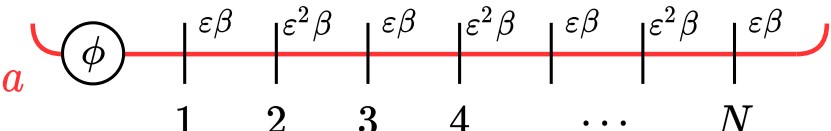

Figure 1: A pictorial illustration of semi-cyclic transfer matrix $\mathbf{T}_a^{\mathrm{sc}}(u, \beta, \phi)$. The auxiliary space is denoted as $a$, while $\phi$ corresponds to the twist matrix. Different values of parameter $\beta$ in semi-cyclic representation are written explicitly next to each physical sites.

$\beta = 0$, the $\ell_2$-dimensional semi-cyclic representation becomes $\ell_2$-dimensional highest weight representation, cf. Appendix A in [30].

Therefore, we define the monodromy matrix for a system with $N$ sites and twist $\phi$

$$\mathbf{M}_s^{\mathrm{sc}}(u, \beta, \phi) = \mathbf{L}_{aN}^{\mathrm{sc}}(u, s, \varepsilon^N \beta) \cdots \mathbf{L}_{aj}^{\mathrm{sc}}(u, s, \varepsilon^j \beta) \cdots \mathbf{L}_{a1}^{\mathrm{sc}}(u, s, \varepsilon \beta) \mathbf{E}_a(\phi), \tag{18}$$

with twist operator $\mathbf{E}_a(\phi) = \sum_{j=0}^{\ell_2 - 1} e^{\mathrm{i}\phi j} |j\rangle_a \langle j|_a$ acting in the auxiliary space. The twist $\phi$ considered here is *commensurate*, cf. (21). As shown in [30], commensurate value of the twist $\phi$ leads to the descendant tower structure and Onsager algebra due to the existence of eigenstates associated with exact (Fabricius-McCoy) strings.

The transfer matrices are defined as

$$\mathbf{T}_s^{\mathrm{sc}}(u, \beta, \phi) = \mathrm{tr}_a \mathbf{M}_s^{\mathrm{sc}}(u, \beta, \phi), \tag{19}$$

depicted in Fig. 1.

Similarly, we define the transfer matrices $\mathbf{T}_s(u, \phi)$ with auxiliary space being $(2s + 1)$-dimensional unitary representation of $\mathcal{U}_q(\mathfrak{sl}_2)$ when $2s \in \mathbb{Z}_{>0}$, see Appendix C.2, i.e.

$$\mathbf{T}_s(u, \phi) = \mathrm{tr}_a \mathbf{M}_s(u, \phi), \quad \mathbf{M}_s(u, \phi) = \mathbf{L}_{aN}(u, s) \cdots \mathbf{L}_{aj}(u, s) \cdots \mathbf{L}_{a1}(u, s, )\mathbf{E}_a(\phi). \tag{20}$$

A detailed discussion of the commutation relations between $\mathbf{T}_s(u, \phi)$ and $\mathbf{T}_s^{\mathrm{sc}}(u, \beta, \phi)$ can be found in [30]. We would like to stress here that the "RLL" relation for the semi-cyclic transfer matrices (17) is satisfied only when the twist $\phi$ is *commensurate* [30] which depends on the parameters $\varepsilon$ and $N$,

$$\begin{aligned} \varepsilon^N = +1 &\Rightarrow \quad \phi = \frac{(2n-2)\pi}{\ell_2}, \\ \varepsilon^N = -1 &\Rightarrow \quad \phi = \frac{(2n-1)\pi}{\ell_2}, \end{aligned} \qquad 1 \le n \le \ell_2, \ n \in \mathbb{N}. \tag{21}$$

It is worth emphasising that $\phi = 0$ (with no twist) is commensurate when $\varepsilon^N = +1$, which has been considered previously in [24–26]. Detailed derivations are shown in [30]. We assume the twist $\phi$ is commensurate when defining the conserved charges below.

In addition to the generating functions for (quasi-)local charges [33–36] associated with $(2s + 1)$-dimensional spin-$s$ representation ($2s \in \mathbb{Z}_{>0}$), cf. the constructions in [30], we define two generating functions for quasilocal Z and Y charges,

$$\mathbf{Z}(u, \phi) = \frac{1}{2\eta} \partial_s \log \mathbf{T}_s^{\mathrm{sc}}(u, \beta, \phi)\Big|_{s=(\ell_2-1)/2, \beta=0}, \tag{22}$$

and

$$\mathbf{Y}(u, \phi) = \frac{1}{2\sinh\eta} \partial_\beta \log \mathbf{T}_s^{\mathrm{sc}}(u, \beta, \phi)\Big|_{s=(\ell_2-1)/2, \beta=0}, \tag{23}$$

where the prefactors are chosen for later convenience in Sec. 2. The significance of the quasilocality of conserved charges in the thermodynamic limit is beyond the scope of this work, and we refer the readers to [37–39] for detailed discussions.

From the "RLL" relation, we could show that these two operators are in involution with themselves respectively [30],

$$[\mathbf{Z}(u,\phi),\mathbf{Z}(v,\phi)] = [\mathbf{Y}(u,\phi),\mathbf{Y}(v,\phi)] = 0, \quad u,v \in \mathbb{C}, \tag{24}$$

and they commute with the Hamiltonian

$$[\mathbf{Z}(u,\phi),\mathbf{H}(\phi)] = [\mathbf{Y}(u,\phi),\mathbf{H}(\phi)] = 0, \quad u \in \mathbb{C}. \tag{25}$$

As for the commutation relations with transfer matrix with auxiliary space as $(2s + 1)$-dimensional spin $s$ representation, cf. Appendix C.2, we have

$$[\mathbf{Z}(u,\phi),\mathbf{T}_s(v,\phi)] = [\mathbf{Y}(u,\phi),\mathbf{T}_s(v,\phi)] = 0, \quad \varepsilon = +1, \quad 2s \in \mathbb{Z}_{>0}, u,v \in \mathbb{C}; \tag{26}$$

$$[\mathbf{Z}(u,\phi),\mathbf{T}_s(v,\phi)] = [\mathbf{Y}(u,\phi),\mathbf{T}_s(v,\phi)] = 0, \varepsilon = -1, s \in \mathbb{Z}_{>0}, u,v \in \mathbb{C},$$
$$[\mathbf{Z}(u,\phi),\mathbf{T}_{s'}(v,\phi)] = \{\mathbf{Y}(u,\phi),\mathbf{T}_{s'}(v,\phi)\} = 0, \varepsilon = -1, s' \in \frac{\mathbb{Z}_{>0}}{2} \setminus \mathbb{Z}_{>0}, u,v \in \mathbb{C}. \tag{27}$$

The anti-commutation between $\mathbf{Y}(u,\phi)$ and $\mathbf{T}_{s'}(v,\phi)$ when $s'$ is a half-integer might seem confusing. The reason is that $\mathbf{Y}(u,\phi)$ anti-commute with momentum operator when $\varepsilon = -1$, that is $\mathbf{Y}(u,\phi)$ is not translational invariant but 2-site translationally invariant which would be clear in Sec. 4.1. This means acting by $\mathbf{Y}(u,\phi)$ would change momentum of a state by $\pi$, resulting in the anti-commutation relation in (27). Moreover, operators $\mathbf{Z}(u,\phi)$ and $\mathbf{Y}(u,\phi)$ do not commute, which is closely related to the conjectured Onsager algebra, cf. Sec. 4.2.

Quasilocal Z and Y charges [36,40] can be obtained by expanding the generating functions at $u = u_0$, i.e.

$$\mathbf{Z}(u,\phi) = \sum_{n=0}^{\infty}(u-u_0)^n \mathbf{Z}_n, \quad \mathbf{Y}(u,\phi) = \sum_{n=0}^{\infty}(u-u_0)^n \mathbf{Y}_n, \tag{28}$$

where

$$\varepsilon = -1 \Rightarrow u_0 = \frac{\eta}{2}; \quad \varepsilon = +1 \Rightarrow u_0 = \frac{\eta - \mathrm{i}\pi}{2}. \tag{29}$$

The reason for the different values of $u_0$ with respect to different $\varepsilon$ values is given in Appendix D.

Generating function $\mathbf{Z}(u,\phi)$ is closely related to the Q operator and 2-parameter transfer matrix for XXZ model at root of unity [30], while generating function $\mathbf{Y}(u,\phi)$ is conjectured to be the creation operator for exact (Fabricius–McCoy) strings in [30].

From (24), all Z or Y charges are in involution with each other,

$$[\mathbf{Z}_m,\mathbf{Z}_n] = [\mathbf{Y}_m,\mathbf{Y}_n] = 0, \quad m,n \in \mathbb{Z}_{\geq 0}, \tag{30}$$

and they can be expressed as

$$\mathbf{Z}_n = \frac{1}{n!}\partial_u^n \mathbf{Z}(u,\phi)|_{u=u_0}, \quad \mathbf{Y}_n = \frac{1}{n!}\partial_u^n \mathbf{Y}(u,\phi)|_{u=u_0}, \tag{31}$$

with the first terms

$$\mathbf{Z}_0 = \mathbf{Z}(u_0,\phi), \quad \mathbf{Y}_0 = \mathbf{Y}(u_0,\phi). \tag{32}$$

# 4 Onsager algebra symmetry in spin-$1/2$ XXZ model at root of unity

XX model (XXZ model with $\Delta = 0$) is a $U(1)$-invariant Hamiltonian that possesses the Onsager algebra symmetry [16], cf. (11). We start this section by discussing the known results in the XX case, which serves as the motivation for the conjectures in Sec. 4.2. We generalise the results in the XX case by conjecturing the existence and properties of hidden Onsager algebra symmetries for XXZ model at arbitrary root of unity.

## 4.1 Example: XX model

We illustrate the relation between Onsager generators and semi-cyclic transfer matrix $\mathbf{T}_s^{sc}(u, \beta, \phi)$ using the example of spin-1/2 XX model. Twist $\phi$ considered here is always commensurate, satisfying (21), namely $\phi \in \{0, \pi\}$ when system size $N$ is even, and $\phi \in \{\pi/2, 3\pi/2\}$ with system size $N$ being odd.

The XX Hamiltonian is [1]

$$\mathbf{H}_{XX} = \sum_{j=1}^{N} \mathbf{h}_j, \quad h_j = \frac{1}{2}\left(\sigma_j^+ \sigma_{j+1}^- + \sigma_j^- \sigma_{j+1}^+\right), \tag{33}$$

where the Onsager generators of algebra $O'$ are

$$\mathbf{Q}_0^0 = \frac{1}{2}\sum_{j=1}^{N} \sigma_j^z = S^z, \quad \mathbf{Q}_0^\pm = 0, \tag{34}$$

$$\mathbf{Q}_1^0 = \frac{i}{2}\sum_{j=1}^{N}\left(\sigma_j^+ \sigma_{j+1}^- - \sigma_j^- \sigma_{j+1}^+\right), \quad \mathbf{Q}_1^+ = -\frac{i}{2}\sum_{j=1}^{N}(-1)^j \sigma_j^+ \sigma_{j+1}^+,$$

$$\mathbf{Q}_1^- = \frac{i}{2}\sum_{j=1}^{N}(-1)^j \sigma_j^- \sigma_{j+1}^-. \tag{35}$$

Using the isomorphism between $O$ and $O'$, we express the Onsager generators of the standard presentation $O$ as

$$\mathbf{Q}_0 = \mathbf{Q}_0^0 + \mathbf{Q}_0^+ + \mathbf{Q}_0^-, \quad \mathbf{Q}_1 = \mathbf{Q}_1^0 + \mathbf{Q}_1^+ + \mathbf{Q}_1^-. \tag{36}$$

The Onsager generators of algebra $O$ satisfy the DG relations

$$\left[\mathbf{Q}_0, \left[\mathbf{Q}_0, \left[\mathbf{Q}_0, \mathbf{Q}_1\right]\right]\right] = \ell_2^2\left[\mathbf{Q}_0, \mathbf{Q}_1\right], \quad \left[\mathbf{Q}_1, \left[\mathbf{Q}_1, \left[\mathbf{Q}_1, \mathbf{Q}_0\right]\right]\right] = \ell_2^2\left[\mathbf{Q}_1, \mathbf{Q}_0\right], \tag{37}$$

with $\ell_2 = 2$.

**Remark.** From the perspective of self-duality, we can equivalently choose

$$\tilde{\mathbf{Q}}_0 = \mathbf{Z}_0 + \mathbf{Y}_0 + \mathbf{Y}_0^\dagger, \quad \tilde{\mathbf{Q}}_1 = \frac{1}{2}\sum_{j=1}^{N} \sigma_j^z, \tag{38}$$

which results in the same Onsager algebra. However, it is natural from the physics of the spin chain to use the parametrisation in (34), (35) and (36). In this parametrisation, we choose the $U(1)$ charge $\mathbf{Q}_0^0$ as the magnetisation $S^z$, and $\mathbf{Q}_m^\pm$ have the physical meaning of changing the states between different magnetisation sectors. The same parametrisation is used for the higher-spin generalisation in Sec. 6.

---

[1]The twist $\phi$ is included in the Hamiltonian through the definition of $\sigma_{N+1}^\pm$, explained in the line below (12).

**Proof of** (37). It is straightforward to check that generators (34) and (35) satisfy

$$\left[\mathbf{Q}_0^0, \mathbf{Q}_1^0\right] = 0, \quad \left[\mathbf{Q}_0^0, \mathbf{Q}_1^\pm\right] = \pm 2\mathbf{Q}_1^\pm. \tag{39}$$

The second equation is a simple consequence that operators $\mathbf{Q}_1^\pm$ change the magnetisation by $\pm 2$. Hence, we have

$$[\mathbf{Q}_0, \mathbf{Q}_1] = 2\left(\mathbf{Q}_1^+ - \mathbf{Q}_1^-\right). \tag{40}$$

Applying the relation above twice, we obtain

$$\left[\mathbf{Q}_0, \left[\mathbf{Q}_0, \left[\mathbf{Q}_0, \mathbf{Q}_1\right]\right]\right] = 4\left[\mathbf{Q}_0, \mathbf{Q}_1\right]. \tag{41}$$

For the second equation in (37), we can compute the commutator explicitly, i.e.

$$\begin{aligned}
\left[\mathbf{Q}_1, \left[\mathbf{Q}_1, \left[\mathbf{Q}_1, \mathbf{Q}_0\right]\right]\right] &= 2\left[\mathbf{Q}_1, \left[\mathbf{Q}_1, \mathbf{Q}_1^- - \mathbf{Q}_1^+\right]\right] \\
&= \left[\mathbf{Q}_1, \sum_{j=1}^N \sigma_j^z - \left(\sigma_j^+ \sigma_{j+1}^z \sigma_{j+2}^- + (-1)^j \sigma_j^+ \sigma_j^z \sigma_{j+2}^+ + \text{h.c.}\right)\right] \\
&= 4\left[\mathbf{Q}_1, \mathbf{Q}_0\right].
\end{aligned} \tag{42}$$

It is straightforward to check that $\mathbf{H}_{\text{XX}}$ is $U(1)$ invariant, i.e. commuting with the Onsager generators of $O'$ (and $O$),

$$\left[\mathbf{H}_{\text{XX}}, \mathbf{Q}_m^r\right] = 0, \quad r \in \{0, +, -\}, \quad m \in \mathbb{Z}. \tag{43}$$

The relation to the canonical generators in Sec. 2 is of the form

$$\mathbf{A}_m^r \to 2\mathbf{Q}_m^r, \quad m \in \mathbb{Z}, \quad r \in \{0, +, -\}. \tag{44}$$

The Onsager generators of $O'$ in terms of local spin operators are given in Appendix E.

What is truly striking here is that the generators $\mathbf{Q}_1^r$ can be identified with the Z and Y charges $\mathbf{Z}_0$ and $\mathbf{Y}_0$, i.e.

$$\mathbf{Q}_1^0 = \mathbf{Z}_0, \quad \mathbf{Q}_1^- = \mathbf{Y}_0, \quad \mathbf{Q}_1^+ = \mathbf{Y}_0^\dagger. \tag{45}$$

**Proof of** (45) We present a simple proof of (45) using direct calculation based on transfer matrices. First, for the Lax operator with two-dimensional semi-cyclic representation ($\ell_2 = 2$), we have

$$\mathbf{L}_{aj}^{\text{sc}}\left(\frac{\mathrm{i}\pi}{4}, \frac{1}{2}, 0\right) = \mathrm{i}\mathbf{P}_{aj}, \tag{46}$$

where permutation operator is defined as

$$\mathbf{P}_{aj} = \frac{1}{2}(\mathbb{1}_a \otimes \mathbb{1}_j + \sigma_a^z \otimes \sigma_j^z) + \sigma_a^+ \otimes \sigma_j^- + \sigma_a^- \otimes \sigma_j^+. \tag{47}$$

The permutation operator $\mathbf{P}_{ab}$ permutes the vector spaces $a$ and $b$, i.e.

$$\mathbf{P}_{ab}\mathbf{X}_a = \mathbf{X}_b\mathbf{P}_{ab}, \quad \mathbf{P}_{ab}\mathbf{X}_b = \mathbf{X}_a\mathbf{P}_{ab}, \tag{48}$$

where $\mathbf{X}_a$ and $\mathbf{X}_b$ are operators acting on the vector spaces $a$ and $b$ respectively.

As for the transfer matrix, we have

$$\mathbf{T}_{1/2}\left(\frac{\eta}{2}, 0, \phi\right) = \mathrm{i}^N \text{tr}_a \mathbf{P}_{aN} \cdots \mathbf{P}_{a2} \mathbf{P}_{a1} \mathbf{E}_a(\phi) = \mathrm{i}^N \mathbf{E}_1(\phi) \mathbf{P}_{12} \mathbf{P}_{23} \cdots \mathbf{P}_{(N-1)N}, \tag{49}$$

making use of the property (48). Similarly, we have

$$
\frac{1}{2\eta} \left. \partial_s \mathbf{T}_s^{\mathrm{sc}}\left(\frac{\eta}{2}, 0, \phi\right)\right|_{s=1/2} = \frac{\mathrm{i}^{N-1}}{4} \sum_{j=1}^{N} \mathbf{E}_1(\phi) \mathbf{P}_{12} \mathbf{P}_{23} \cdots \mathbf{P}_{(j-2)(j-1)}
$$
$$
(\sigma_{j+1}^z \otimes \mathbb{1}_j - \mathbb{1}_{j+1} \otimes \sigma_j^z) \mathbf{P}_{(j-1)(j+1)} \cdots \mathbf{P}_{(N-1)N},
$$

(50)

and

$$
\frac{1}{2\sinh\eta} \left. \partial_\beta \mathbf{T}_{1/2}^{\mathrm{sc}}\left(\frac{\eta}{2}, \beta, \phi\right)\right|_{\beta=0} = \frac{\mathrm{i}^{N-1}}{2} \sum_{j=1}^{N} (-1)^j \mathbf{E}_1(\phi) \mathbf{P}_{12} \mathbf{P}_{23} \cdots \mathbf{P}_{(j-2)(j-1)}
$$
$$
(\sigma_a^- \otimes \sigma_j^-) \mathbf{P}_{(j-1)(j+1)} \cdots \mathbf{P}_{(N-1)N}.
$$

(51)

Combining with (49), we prove the two formulae in (45).

This relation has been pointed out in [16], and all operators $\mathbf{Q}_n^r$, $\mathbf{Z}_n$ and $\mathbf{Y}_n$ can be written in terms of bilinear fermion operators, reflecting the free fermion nature of XX model. One of the consequences is that there exists a closure condition for Onsager generators in XX model, namely

$$
\mathbf{Q}_{n+2N}^r = \mathbf{Q}_n^r, \quad \forall n \in \mathbb{Z}.
$$

(52)

The discussion of the physical meaning of the closure condition is postponed to Sec. 4.3. Since there are infinitely many $\mathbf{Q}_n^r$, $\mathbf{Z}_n$ and $\mathbf{Y}_n$, the relation between all of them needs to be elucidated. In fact, all operators can be obtained recursively, and the first few read

$$
\mathbf{Z}_1 = \frac{1}{1!}\left(\mathbf{Q}_2^0 - \mathbf{Q}_0^0\right), \quad \mathbf{Z}_2 = \frac{1}{2!}\left(2\mathbf{Q}_3^0 - 2\mathbf{Q}_1^0\right), \quad \mathbf{Z}_3 = \frac{1}{3!}\left(6\mathbf{Q}_4^0 - 8\mathbf{Q}_2^0 + 2\mathbf{Q}_0^0\right),
$$
$$
\mathbf{Y}_1 = \frac{1}{1!}\left(\mathbf{Q}_2^- - \mathbf{Q}_0^-\right), \quad \mathbf{Y}_2 = \frac{1}{2!}\left(2\mathbf{Q}_3^- - 2\mathbf{Q}_1^-\right), \quad \mathbf{Y}_3 = \frac{1}{3!}\left(6\mathbf{Q}_4^- - 8\mathbf{Q}_2^- + 2\mathbf{Q}_0^-\right),
$$

(53)

revealing a deep connection between the Onsager generators $\mathbf{Q}_n^r$ and conserved charges $\mathbf{Z}_n$, $\mathbf{Y}_n$. Relations between higher-order terms can be obtained recursively.

**Derivation of** (53)  Instead of using the Lax operator directly which becomes cumbersome when considering higher-order Z and Y charges, we present a derivation for the higher-order Z and Y charges in terms of Onsager generators using boost operator approach [41–43]. The boost operator in the XX models are defined as

$$
\mathcal{B}^{\mathrm{XX}} = \sum_j \mathcal{B}_j^{\mathrm{XX}}, \quad \mathcal{B}_j^{\mathrm{XX}} = \left(j + \frac{1}{2}\right)\frac{1}{2}\left(\sigma_j^+ \sigma_{j+1}^- + \sigma_j^- \sigma_{j+1}^+\right).
$$

(54)

The higher-order Z and Y charges are generated by commuting with boost operator (54),

$$
\mathbf{Z}_m = \frac{\mathrm{i}}{m}\left[\mathbf{Z}_{m-1}, \mathcal{B}^{\mathrm{XX}}\right], \quad \mathbf{Y}_m = \frac{\mathrm{i}}{m}\left[\mathbf{Y}_{m-1}, \mathcal{B}^{\mathrm{XX}}\right],
$$

(55)

with $m \in \mathbb{Z}_{>0}$. As for the validity of (55), we show a sketch of a proof that can be generalised to the spin-$s$ cases with $\ell_2 = 2s + 1$ in Appendix F. Bearing this postulation in mind, we start with a lemma,

$$
\mathrm{i}\left[\mathbf{Q}_m^r, \mathcal{B}^{\mathrm{XX}}\right] = m\left(\mathbf{Q}_{m+1}^r - \mathbf{Q}_m^r\right), \quad r \in \{0, +, -\},
$$

(56)

where $\mathbf{Q}_m^r$ are given in Appendix E. The proof of the lemma is straightforward after telescoping the series. We demonstrate the proofs of Lemma (56) in Appendix G.

Since we have already known that $\mathbf{Z}_0 = \mathbf{Q}_1^0$ in (45), we obtain higher-order relations by applying (56) recursively,

$$
\begin{aligned}
\mathbf{Z}_1 &= \frac{i}{1}\left[\mathbf{Z}_0, \mathcal{B}^{\mathrm{XX}}\right] = \mathbf{Q}_2^0 - \mathbf{Q}_0^0, \\
\mathbf{Z}_2 &= \frac{i}{2}\left[\mathbf{Z}_1, \mathcal{B}^{\mathrm{XX}}\right] = \mathbf{Q}_3^0 - \mathbf{Q}_1^0 = \frac{1}{2!}\left(2\mathbf{Q}_3^0 - 2\mathbf{Q}_1^0\right), \\
\mathbf{Z}_3 &= \frac{i}{3}\left[\mathbf{Z}_2, \mathcal{B}^{\mathrm{XX}}\right] = \mathbf{Q}_4^0 - \frac{4}{3}\mathbf{Q}_2^0 + \frac{1}{3}\mathbf{Q}_0^0 = \frac{1}{3!}\left(6\mathbf{Q}_4^0 - 8\mathbf{Q}_2^0 + 2\mathbf{Q}_0^0\right).
\end{aligned}
\tag{57}
$$

Similar expressions for Y charges are obtained analogously.

Moreover, we can write down a recursive expressions for the expansion of $\mathbf{Z}_n$ and $\mathbf{Y}_n$ in terms of Onsager generators $\mathbf{Q}_m^r$, i.e.

$$
\begin{aligned}
\mathbf{Z}_n &= \sum_{j=0}^{\lfloor (n+1)/2 \rfloor} c_j^n \mathbf{Q}_{(n+1)-2j}^0, \\
\mathbf{Y}_n &= \sum_{j=0}^{\lfloor (n+1)/2 \rfloor} c_j^n \mathbf{Q}_{(n+1)-2j}^-, \quad n \in \mathbb{Z},
\end{aligned}
\tag{58}
$$

where the coefficients are

$$
c_{m+1}^m = c_2^1 = 1, \quad c_x^{m+1} = \frac{x-1}{m+1}c_{x-1}^m - \frac{x+1}{m+1}c_{x+1}^m.
\tag{59}
$$

## 4.2 Conjectures on hidden Onsager algebra symmetries in spin-$1/2$ XXZ models at root of unity

Motivated by the exact correspondence between the Onsager generators of $O'$ $\mathbf{Q}_n^r$ and conserved charges (Z and Y charges) in the XX case ($q = \exp(i\pi/2) = i$), cf. (45) and (53), it is natural to generalise similar relations for the spin-1/2 XXZ model at arbitrary root of unity ($q = \exp(i\pi\ell_1/\ell_2)$), despite that the operators $\mathbf{Q}_n^r$, $Z_n$ and $Y_n$ are no longer able to be expressed in local densities but quasilocal ones [33–36]. Another motivation to the following conjectures is that the structure of descendant towers and exact (Fabricius–McCoy) strings are of no difference between XX model and XXZ model at other root of unity. After numerically verifying the relations, we are able to compose conjectures for the existence of the hidden Onsager algebra symmetry in spin-1/2 XXZ model at arbitrary root of unity, which are as follows:

**Conjecture I:**
There exists a hidden Onsager algebra symmetry in spin-1/2 XXZ spin chain at root of unity with commensurate twist (21). Spin-1/2 XXZ models at root of unity with commensurate twist (21) are $U(1)$ invariant, i.e.

$$
\left[\mathbf{H}(\phi), \mathbf{Q}_m^r\right] = 0, \quad r \in \{0, +, -\}, \quad m \in \mathbb{Z}.
\tag{60}
$$

The generators of the hidden Onsager algebra in the presentation $O'$ are obtained by means of (44), i.e.

$$
\mathbf{Q}_0^0 = \frac{1}{2}\sum_{j=1}^{N} \sigma^z, \quad \mathbf{Q}_0^{\pm} = 0,
\tag{61}
$$

$$
\begin{aligned}
\mathbf{Q}_1^0 &= \mathbf{Z}_0 = \frac{1}{2\eta}\partial_s \log \mathbf{T}_s^{\mathrm{sc}}(u, \beta, \phi)\Big|_{s=(\ell_2-1)/2, u=u_0, \beta=0}, \\
\mathbf{Q}_1^- &= \mathbf{Y}_0 = \frac{1}{2\sinh\eta}\partial_\beta \log \mathbf{T}_s^{\mathrm{sc}}(u, \beta, \phi)\Big|_{s=(\ell_2-1)/2, u=u_0, \beta=0} = \left(\mathbf{Q}_1^+\right)^\dagger.
\end{aligned}
\tag{62}
$$

We conjecture the identification between the canonical generators $\mathbf{A}_m^r$ and the generators $\mathbf{Q}_m^r$ for arbitrary root of unity to be of the form

$$\mathbf{A}_m^r \rightarrow \frac{\ell_2}{4}\mathbf{Q}_m^r, \quad m \in \mathbb{Z}, \quad r \in \{0, +, -\}. \tag{63}$$

From the relation (36), the generators of the presentation $O\,\mathbf{Q}_m$ are conjectured to satisfy the DG relation

$$\Big[\mathbf{Q}_0,\big[\mathbf{Q}_0,[\mathbf{Q}_0,\mathbf{Q}_1]\big]\Big] = \ell_2^2[\mathbf{Q}_0,\mathbf{Q}_1], \quad \Big[\mathbf{Q}_1,\big[\mathbf{Q}_1,[\mathbf{Q}_1,\mathbf{Q}_0]\big]\Big] = \ell_2^2[\mathbf{Q}_1,\mathbf{Q}_0], \tag{64}$$

with any $\ell_2 \in \mathbb{Z}_{>0}$. Notice that the definition of semi-cyclic transfer matrix $\mathbf{T}_s^{sc}(u, \beta, \phi)$ depends on the root of unity through $\varepsilon = \pm 1$.

Similar to (53), the higher-order Onsager generators are conjectured to be related to the higher-order Z and Y charges.

**Conjecture II:**
In general, the higher-order Z and Y charges are functions of the higher-order Onsager generators such that

$$\mathbf{Z}_n = \left(\frac{\ell_2}{2}\right)^n \sum_{j=0}^{\lfloor(n+1)/2\rfloor} c_j^n \mathbf{Q}_{(n+1)-2j}^0,$$
$$\mathbf{Y}_n = \left(\frac{\ell_2}{2}\right)^n \sum_{j=0}^{\lfloor(n+1)/2\rfloor} c_j^n \mathbf{Q}_{(n+1)-2j}^-, \quad n \in \mathbb{Z}, \tag{65}$$

where $c_j^n \in \mathbb{N}$. The first three terms $\mathbf{Z}_n$ and $\mathbf{Y}_n$, $n \in \{1, 2, 3\}$ can be expressed as

$$
\begin{aligned}
\mathbf{Z}_1 &= \frac{1}{1!}\frac{\ell_2}{2}\left(\mathbf{Q}_2^0 - \mathbf{Q}_0^0\right), \quad \mathbf{Z}_2 = \frac{1}{2!}\left(\frac{\ell_2}{2}\right)^2\left(2\mathbf{Q}_3^0 - 2\mathbf{Q}_1^0\right), \\
\mathbf{Z}_3 &= \frac{1}{3!}\left(\frac{\ell_2}{2}\right)^3\left(6\mathbf{Q}_4^0 - 8\mathbf{Q}_2^0 + 2\mathbf{Q}_0^0\right), \\
\mathbf{Y}_1 &= \frac{1}{1!}\frac{\ell_2}{2}\left(\mathbf{Q}_2^- - \mathbf{Q}_0^-\right), \quad \mathbf{Y}_2 = \frac{1}{2!}\left(\frac{\ell_2}{2}\right)^2\left(2\mathbf{Q}_3^- - 2\mathbf{Q}_1^-\right), \\
\mathbf{Y}_3 &= \frac{1}{3!}\left(\frac{\ell_2}{2}\right)^3\left(6\mathbf{Q}_4^- - 8\mathbf{Q}_2^- + 2\mathbf{Q}_0^-\right).
\end{aligned}
\tag{66}
$$

Conjecture I (62) and Conjecture II (66) are proven for the case of XX model ($\ell_2 = 2$), illustrated in Sec. 4.1. Conjectures I and II, cf. (62) (as well as the identification between the canonical generators $\mathbf{A}_m^r$ and the generators $\mathbf{Q}_m^r$ in (63)) and (66) have been verified numerically for cases whose roots of unity satisfy $\ell_2 = 3, 4, 5$ and all permitted values of $\ell_1$ with system size $N$ up to 12. The numerical evidence is convincing that Conjectures I and II are true for arbitrary root of unity value of the anisotropy and system size.

## 4.3 Closure condition: free v.s. interacting

Let us assume that the conjectures above are true. One might wonder the question about the physical difference between XX model and XXZ model at root of unity other than $\exp(i\pi/2)$. On the one hand, they all possess Onsager algebra symmetries, which are identical on the level of algebraic structure; on the other hand, XX model permits a free fermionic description [16], while XXZ model at other root of unity does not, due to its intrinsically interacting nature [20].

The explicit forms of the Onsager generators of $O$ $\mathbf{Q}_m$ of different physical models in consideration are regarded as different representations of the Onsager algebra. Even though the algebraic structure of those generators for different models (1) is identical, the generators for different models still have different properties. All the Onsager generators of $O$ (and $O'$) for XX model are bilinear in fermionic operators [16]. This can be seen by performing Jordan–Wigner transformation for (E.1) and (E.2). It implies that for XXZ model at root of unity, the representation associated with $q = \mathrm{i}$ has the free fermionic behaviour, resulting in the closure condition (52) [16, 17]. For other roots of unity $q = \exp\left(\mathrm{i}\pi\frac{\ell_1}{\ell_2}\right) \neq \exp(\mathrm{i}\pi/2)$, the closure condition is no longer satisfied, since these models are interacting. This observation suggests that we cannot transform all the nice properties and methods used for XX model to XXZ models at other roots of unity.

**Remark.** Historically speaking, Onsager solved the partition function of two-dimensional Ising model using both commutation relations between Onsager generators of $O$ (1) and the closure condition (52) [1]. Both of them are crucial to the exact solutions.

## 5 Spin-$1$ case: transfer matrix fusion

We can generalise the construction in the spin-1/2 case by considering the spin-1 generalisation of XXZ model (12), i.e. ZF model [44]. The spin-1 ZF model is the Hamiltonian limit to the Izergin-Korepin 19-vertex model [45], leading to the construction of transfer matrix. The $N$-site quasi-periodic ZF model reads

$$
\begin{aligned}
\mathsf{H}_{\mathrm{ZF}}(\eta, \phi) = \sum_{j=1}^{N}\Big( & \Big[ \mathsf{S}_j^x \mathsf{S}_{j+1}^x + \mathsf{S}_j^y \mathsf{S}_{j+1}^y + \cosh(2\eta)\mathsf{S}_j^z \mathsf{S}_{j+1}^z \Big] \\
& + 2\Big[ \left(\mathsf{S}_j^x\right)^2 + \left(\mathsf{S}_j^y\right)^2 + \cosh(2\eta)\left(\mathsf{S}_j^x\right)^2 \Big] - \sum_{a,b} \mathsf{A}_{ab}(\eta)\mathsf{S}_j^a \mathsf{S}_j^b \mathsf{S}_{j+1}^a \mathsf{S}_{j+1}^b \Big),
\end{aligned}
\tag{67}
$$

where coefficients $\mathsf{A}_{ab} = \mathsf{A}_{ba}$ take the values of

$$
\mathsf{A}_{xx} = \mathsf{A}_{yy} = 1, \ \mathsf{A}_{zz} = \cosh(2\eta), \ \mathsf{A}_{xy} = 1, \ \mathsf{A}_{xz} = \mathsf{A}_{yz} = 2\cosh\eta - 1.
\tag{68}
$$

The spin-1 operators are

$$
\begin{aligned}
\mathsf{S}^x = \frac{1}{\sqrt{2}}\begin{pmatrix} 0 & 1 & 0 \\ 1 & 0 & 1 \\ 0 & 1 & 0 \end{pmatrix}, \quad \mathsf{S}^y = \frac{1}{\sqrt{2}}\begin{pmatrix} 0 & -\mathrm{i} & 0 \\ \mathrm{i} & 0 & -\mathrm{i} \\ 0 & \mathrm{i} & 0 \end{pmatrix}, \\
\mathsf{S}^z = \begin{pmatrix} 1 & 0 & 0 \\ 0 & 0 & 0 \\ 0 & 0 & -1 \end{pmatrix}, \quad \mathsf{S}^{\pm} = \mathsf{S}^x \pm \mathrm{i}\mathsf{S}^-,
\end{aligned}
\tag{69}
$$

and $\mathsf{S}_j^\alpha = \mathbb{1}_3^{\otimes(j-1)} \otimes \mathsf{S}^\alpha \otimes \mathbb{1}_3^{\otimes(N-j)}$. The twist is encoded in the relation $\mathsf{S}_{N+1}^\pm = e^{\pm\mathrm{i}\phi}\mathsf{S}_1^\pm$.

The anisotropy parameter $\eta$ can be re-parametrised in terms of $q = \exp\eta$. At root of unity value $q = \exp\left(\mathrm{i}\pi\frac{\ell_1}{\ell_2}\right)$ with $\ell_1$ and $\ell_2$ being coprimes, we define parameter $\varepsilon = q^{\ell_2} = \pm 1$.

When $\eta = 0$, i.e. the isotropic limit, $\mathsf{H}_{\mathrm{ZF}} \propto \sum_j \vec{\mathsf{S}}_j \cdot \vec{\mathsf{S}}_{j+1} - (\vec{\mathsf{S}}_j \cdot \vec{\mathsf{S}}_{j+1})^2$. This is known in the literature as spin-1 Takhtajan–Babujian model [46–48], a spin-1 generalisation of spin-1/2 XXX model.

Spin-1 ZF model is integrable in the same way as spin-1/2 XXZ model, and its Lax operator can be obtained through the transfer matrix fusion relation [49, 50]. What has not been

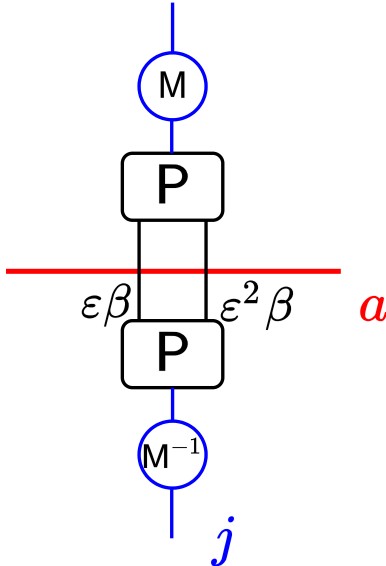

Figure 2: The semi-cyclic Lax operator of ZF model is expressed as a fusion of two Lax operators of spin-1/2 XXZ model, cf. (70). Blue lines correspond to physical Hilbert space with spin-1 representation of $\mathfrak{su}_2$ algebra, which are obtained through the fusion of two spin-1/2 representations denoted as black lines.

obtained previously is the semi-cyclic transfer matrix for ZF model with $\varepsilon = -1$. Here we perform the transfer matrix fusion procedure for the semi-cyclic transfer matrix with arbitrary $\varepsilon = \pm 1$ at root of unity.

The fusion procedure for Lax operator can be described pictorially in Fig. 2.

Equivalently, in terms of formulae, we express the semi-cyclic Lax operator $\mathsf{L}^{sc}_{aj}$ as

$$
\mathsf{L}^{sc}_{aj}(u,s,\beta) = \frac{1}{\sinh^2 \eta} (\mathbb{1}_a \otimes \mathsf{M}_j)(\mathbb{1}_a \otimes \mathsf{P}_{mn})\mathbf{L}^{sc}_{am}(u - \eta/2, s, \varepsilon\beta) \tag{70}
$$
$$
\mathbf{L}^{sc}_{an}(u + \eta/2, s, \varepsilon^2\beta)(\mathbb{1}_a \otimes \mathsf{M}^{-1}_j)(\mathbb{1}_a \otimes \mathsf{P}^{\mathrm{T}}_{mn}),
$$

where P operator projects operators in Hilbert space $mn$ (spin-$\frac{1}{2} \otimes \frac{1}{2}$) to operators in Hilbert space $j$ (spin-1) and M operator fixes the normalisation of Lax operator, i.e.

$$
\mathsf{P}_{mn} = \begin{pmatrix} 1 & 0 & 0 & 0 \\ 0 & \frac{1}{\sqrt{2}} & \frac{1}{\sqrt{2}} & 0 \\ 0 & 0 & 0 & 1 \end{pmatrix}_{mn}, \quad \mathsf{M}_j = \begin{pmatrix} 1 & 0 & 0 \\ 0 & \frac{\sqrt{[2]}}{\sqrt{2}} & 0 \\ 0 & 0 & 1 \end{pmatrix}_j. \tag{71}
$$

Here $q$-number is defined as $[x] = (q^x - q^{-x})/(q - q^{-1})$.

When $\varepsilon = +1$, (70) reproduces the known result [16],

$$
\mathsf{L}^{sc}_{aj}(u,s,\beta) = \begin{pmatrix} [\frac{u}{\eta} - \frac{1}{2} + \mathbf{S}^z_a][\frac{u}{\eta} + \frac{1}{2} + \mathbf{S}^z_a] & \mathbf{S}^-_a[\frac{u}{\eta} - \frac{1}{2} + \mathbf{S}^z_a] & (\mathbf{S}^-_a)^2 \\ \mathbf{S}^+_a[\frac{u}{\eta} + \frac{1}{2} + \mathbf{S}^z_a] & \mathbf{S}^+_a\mathbf{S}^-_a + [\frac{u}{\eta} + \frac{1}{2} + \mathbf{S}^z_a][\frac{u}{\eta} - \frac{1}{2} - \mathbf{S}^z_a] & \mathbf{S}^-_a[\frac{u}{\eta} - \frac{3}{2} + \mathbf{S}^z_a] \\ (\mathbf{S}^+_a)^2 & \mathbf{S}^+_a[\frac{u}{\eta} - \frac{1}{2} - \mathbf{S}^z_a] & [\frac{u}{\eta} + \frac{1}{2} - \mathbf{S}^z_a][\frac{u}{\eta} - \frac{1}{2} - \mathbf{S}^z_a] \end{pmatrix}_j. \tag{72}
$$

However, when $\varepsilon = -1$, relation (72) no longer holds. One should use (70) instead.

It is straightforward to show the RLL relation for the semi-cyclic Lax operator $\mathsf{L}^{sc}_{aj}(u,s,\beta)$,

$$
\mathsf{R}^{sc}_{aj}(u - v, s, \beta)\mathsf{L}^{sc}_{ak}(u, s, \beta)\mathsf{L}_{jk}(v) = \mathsf{L}_{jk}(v)\mathsf{L}^{sc}_{ak}(u, s, \beta)\mathsf{R}^{sc}_{am}(u - v, s, \beta), \tag{73}
$$

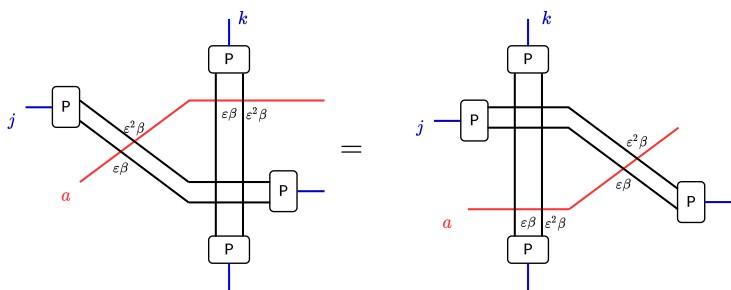

Figure 3: A pictorial proof of (73). We omit the M matrix parts in the Lax operator (70), which do not change the result.

where $\beta \in \mathbb{C}$, and R matrix $\mathsf{R}^{\mathrm{sc}}_{aj}(u) = \mathsf{L}^{\mathrm{sc}}_{aj}(u + \eta/2)$. The proof is constructive and it is shown in Fig. 3.

Via the fusion in Fig. 2, spin-1 ZF model can be seen as a fused spin-1/2 XXZ model with even site. Hence, only the first condition of (21) remains for spin-1 ZF model. We have

$$\varepsilon = \pm 1 \Rightarrow \phi = \frac{(2n-2)\pi}{\ell_2}, \quad 1 \leq n \leq \ell_2, \; n \in \mathbb{N}. \tag{74}$$

We define the monodromy matrix accordingly

$$\mathsf{M}^{\mathrm{sc}}_s(u, \beta, \phi) = \mathsf{L}^{\mathrm{sc}}_{aN}(u, s, \beta) \cdots \mathsf{L}^{\mathrm{sc}}_{aj}(u, s, \beta) \cdots \mathsf{L}^{\mathrm{sc}}_{a1}(u, s, \beta)\mathbf{E}_a(\phi), \tag{75}$$

and semi-cyclic transfer matrix becomes

$$\mathsf{T}^{\mathrm{sc}}_s(u, \beta, \phi) = \mathrm{tr}_a \mathsf{M}^{\mathrm{sc}}_s(u, \beta, \phi). \tag{76}$$

Precisely the same as the spin-1/2 case, we define two generating functions for quasilocal Z and Y charges [50],

$$\mathsf{Z}(u, \phi) = \frac{1}{2\eta} \left. \partial_s \log \mathsf{T}^{\mathrm{sc}}_s(u, \beta, \phi) \right|_{s=(\ell_2-1)/2, \beta=0}, \tag{77}$$

and

$$\mathsf{Y}(u, \phi) = \frac{1}{2 \sinh \eta} \left. \partial_\beta \log \mathsf{T}^{\mathrm{sc}}_s(u, \beta, \phi) \right|_{s=(\ell_2-1)/2, \beta=0}. \tag{78}$$

More importantly, from the RLL relation (73), $\mathsf{Z}(u, \phi)$ and $\mathsf{Y}(u, \phi)$ satisfy identical relations as their counterparts in spin-1/2 case, cf. Eqs. (24), (26) and (27). We shall not recite them again here.

We expand the generating functions at spectral parameter $u_0 = \eta/2$, i.e.

$$\mathsf{Z}(u, \phi) = \sum_{n=0}^{\infty} (u - u_0)^n \mathsf{Z}_n, \quad \mathsf{Y}(u, \phi) = \sum_{n=0}^{\infty} (u - u_0)^n \mathsf{Y}_n. \tag{79}$$

Identical to the spin-1/2 counterparts, Z or Y charges in spin-1 case are in involution with each other respectively,

$$[\mathsf{Z}_m, \mathsf{Z}_n] = [\mathsf{Y}_m, \mathsf{Y}_n] = 0, \quad m, n \in \mathbb{Z}_{\geq 0}. \tag{80}$$

They are expressed as

$$\mathsf{Z}_n = \frac{1}{n!} \partial^n_u \mathsf{Z}(u, \phi)|_{u=u_0}, \quad \mathsf{Y}_n = \frac{1}{n!} \partial^n_u \mathsf{Y}(u, \phi)|_{u=u_0}. \tag{81}$$

The first terms are

$$Z_0 = Z(u_0, \phi), \quad Y_0 = Y(u_0, \phi), \tag{82}$$

which are important when constructing Onsager generators, cf. Sec. 6.1.

It has been shown in [50] that for arbitrary root of unity, $Z(u, \phi)$ and $Y(u, \phi)$ are quasilocal and thus $Z_m$ and $Y_m$ are quasilocal too. One special case is at $\eta = \frac{i\pi}{3}$ or $\frac{2i\pi}{3}$, with $Z_m$ and $Y_m$ written in terms of local operators [16], which we will exploit in Sec. 6.1.

# 6 Onsager algebra symmetry in spin-1 ZF model at root of unity

We proceed in analogue to the spin-1/2 case. We start with an example of spin-1 ZF model with $\eta = i\pi/3$, which possesses the Onsager algebra symmetry in terms of operators with local density in spite of its interacting nature. The construction below works for the case with $\eta = 2i\pi/3$ too, since the Hamiltonians with $\eta$ and $\pi - \eta$ are mapped to each other through a unitary transformation [50]. This model is also known as spin-1 $U(1)$-invariant clock model in [16]. Furthermore, we compose the conjectures for the hidden Onsager algebra symmetries for spin-1 ZF model at arbitrary root of unity values of the anisotropy, similar to the spin-1/2 case.

## 6.1 Example: spin-1 $U(1)$-invariant clock model

We concentrate on the case of spin-1 $U(1)$-invariant clock model in this section. We introduce three additional operators for later convenience,

$$\tau = \begin{pmatrix} 1 & 0 & 0 \\ 0 & \omega & 0 \\ 0 & 0 & \omega^2 \end{pmatrix}, \quad \mathcal{S}^+ = \sum_{k=1}^{2} \mathbf{e}^{k,k+1} = \begin{pmatrix} 0 & 1 & 0 \\ 0 & 0 & 1 \\ 0 & 0 & 0 \end{pmatrix} = \left(\mathcal{S}^-\right)^\dagger, \tag{83}$$

where $\omega = \exp(2i\pi/3)$ and matrix $(\mathbf{e}^{ab})_{cd} = \delta_c^a \delta_d^b$.

We rewrite the Hamiltonian of spin-1 $U(1)$-invariant clock model, i.e. ZF Hamiltonian with $\eta = \frac{i\pi}{3}$ in terms of operators defined above up to a constant, i.e.

$$H_{\text{ZF}}(\phi) = -\sum_{j=1}^{N}\sum_{a=1}^{2}\left[(-1)^a\left(\mathcal{S}_j^-\mathcal{S}_{j+1}^+\right)^a + (-1)^a\left(\mathcal{S}_j^+\mathcal{S}_{j+1}^-\right)^a\right. $$
$$\left. + \frac{3-2a}{3}e^{i\pi a/3}\tau_j^a - \frac{2}{3}\right] = \sum_{j=1}^{N}h_j, \quad \eta = \frac{i\pi}{3}, \tag{84}$$

$$h_j = \mathcal{S}_j^-\mathcal{S}_{j+1}^+ + \mathcal{S}_j^+\mathcal{S}_{j+1}^- - \left(\mathcal{S}_j^-\mathcal{S}_{j+1}^+\right)^2 - \left(\mathcal{S}_j^+\mathcal{S}_{j+1}^-\right)^2 - \frac{1}{2}\left(S_j^z\right)^2 - \frac{1}{2}\left(S_{j+1}^z\right)^2, \tag{85}$$

where $\mathcal{S}_j^\pm = \mathbb{1}_3^{\otimes(j-1)} \otimes \mathcal{S}^\pm \otimes \mathbb{1}_3^{\otimes(N-j)}$, $\tau_j = \mathbb{1}_3^{\otimes(j-1)} \otimes \tau \otimes \mathbb{1}_3^{\otimes(N-j)}$ and $\mathcal{S}_{L+1}^\pm = e^{i\phi}\mathcal{S}_1^\pm$. The Hamiltonian (84) is the same as Eq. (2.9) in [16] up to a unitary gauge transformation, and it can be seen as a generalisation of the spin-1/2 XX model [16]. The local term $h_j$ is expressed in a symmetric way with respect to the $j$th and $(j+1)$th terms in order to get the correct boost operator, cf. (93).

As explained in [16], the Hamiltonian (84) is a $U(1)$ invariant Hamiltonian that possesses the Onsager algebra symmetry (11),

$$\left[H_{\text{ZF}}(\phi), Q_m^r\right] = 0, \quad r \in \{0, +, -\}, \quad m \in \mathbf{Z}. \tag{86}$$

The Onsager generators of algebra $O'$ in terms of operators (83) are

$$Q_0^0 = \sum_{j=1}^{N} S_j^z, \quad Q_0^{\pm} = 0, \tag{87}$$

$$Q_1^0 = \sum_{j=1}^{N} \sum_{a=1}^{2} \frac{\omega^a}{1-\omega^{-a}} \left[ \left( \mathcal{S}_j^- \mathcal{S}_{j+1}^+ \right)^a - \left( \mathcal{S}_j^+ \mathcal{S}_{j+1}^- \right)^a \right], \quad Q_1^+ = \left( Q_1^- \right)^\dagger,$$

$$Q_1^- = \sum_{j=1}^{N} \sum_{a=1}^{2} \frac{\omega^a}{1-\omega^{-a}} \left( \mathcal{S}_j^- \right)^a \left( \mathcal{S}_{j+1}^- \right)^{3-a}, \quad \omega = e^{2i\pi/3}. \tag{88}$$

The relation to Onsager generators of algebra $O$ becomes

$$Q_0 = Q_0^0 + Q_0^+ + Q_0^-, \quad Q_1 = Q_1^0 + Q_1^+ + Q_1^-. \tag{89}$$

The Onsager generators of $O$ satisfy the DG relations, i.e.

$$\left[ Q_0, \left[ Q_0, \left[ Q_0, Q_1 \right] \right] \right] = \ell_2^2 [Q_0, Q_1], \quad \left[ Q_1, \left[ Q_1, \left[ Q_1, Q_0 \right] \right] \right] = \ell_2^2 [Q_1, Q_0], \tag{90}$$

with $\ell_2 = 3$ in this case. Higher-order generators can be obtained through applying relations (4) and (5) recursively. This Hamiltonian (84) is special, since the Onsager generators are in fact local, instead of quasilocal in the generic cases.

Moreover, identical to the example in Sec. 4.1, the Onsager generators are expressed in terms of Z and Y charges, which are able to be written in local densities when $\eta = i\pi/3$,

$$Q_1^0 = Z_0, \quad Q_1^- = Y_0, \quad Q_1^+ = Y_0^\dagger. \tag{91}$$

This relation is derived again using the explicit form of Lax operator, which is similar to the derivation in Sec. 4.1.

This analogue are extended further. We find precisely the same relation as in spin-1/2 cases with finite system sizes $N \sim 10$, cf. (66),

$$Z_1 = \frac{\ell_2}{2} \frac{1}{1!} \left( Q_2^0 - Q_0^0 \right), \quad Z_2 = \left( \frac{\ell_2}{2} \right)^2 \frac{1}{2!} \left( 2Q_3^0 - Q_1^0 \right),$$

$$Z_3 = \left( \frac{\ell_2}{2} \right)^3 \frac{1}{3!} \left( 6Q_4^0 - 8Q_2^0 + 2Q_0^0 \right),$$

$$Y_1 = \frac{\ell_2}{2} \frac{1}{1!} \left( Q_2^- - Q_0^- \right), \quad Y_2 = \left( \frac{\ell_2}{2} \right)^2 \frac{1}{2!} \left( 2Q_3^- - Q_1^- \right), \tag{92}$$

$$Y_3 = \left( \frac{\ell_2}{2} \right)^3 \frac{1}{3!} \left( 6Q_4^- - 8Q_2^- + 2Q_0^- \right),$$

with $\ell_2 = 3$.

**Derivation of** (92)  Similar to the XX case, we derive the higher-order Z and Y charges using boost operator approach. However, the model is interacting and more terms are involved when constructing the higher-order charges. Here we sketch the method and present explicit results for $Z_1$ and $Y_1$ in Appendix H. The boost operator for spin-1 ZF model with $\eta = i\pi/3$ is of form

$$\mathcal{B}^{\mathrm{ZF}} = \sum_j \mathcal{B}_j^{\mathrm{ZF}} = \frac{i}{\sinh \eta} \sum_j \left( j + \frac{1}{2} \right) h_j. \tag{93}$$

After regrouping the elements, we have

$$
\begin{aligned}
\mathcal{B}_j^{\mathrm{ZF}} = \frac{2}{\sqrt{3}} \Big[ & \Big(j+\frac{1}{2}\Big)\mathcal{S}_j^+\mathcal{S}_{j+1}^- + \Big(j+\frac{1}{2}\Big)\mathcal{S}_j^-\mathcal{S}_{j+1}^+ - \Big(j+\frac{1}{2}\Big)\big(\mathcal{S}_j^+\mathcal{S}_{j+1}^-\big)^2 \\
& - \Big(j+\frac{1}{2}\Big)\big(\mathcal{S}_j^-\mathcal{S}_{j+1}^+\big)^2 - j\big(\mathsf{S}_j^z\big)^2 \Big].
\end{aligned}
\tag{94}
$$

Higher-order Z and Y charges are postulated to be

$$
\mathsf{Z}_m = \frac{\mathrm{i}}{m}\big[\mathsf{Z}_{m-1},\mathcal{B}^{\mathrm{ZF}}\big], \quad \mathsf{Y}_m = \frac{\mathrm{i}}{m}\big[\mathsf{Y}_{m-1},\mathcal{B}^{\mathrm{ZF}}\big],
\tag{95}
$$

identical to the XX case.

Due to the interacting nature, cf. Sec. 4.3, the closure condition (52) is absent for spin-1 ZF model with $\eta = \mathrm{i}\pi/3$, or equivalently spin-1 $U(1)$-invariant clock model [16]. Hence, the Onsager generators of $O\,\mathsf{Q}_m$ obtained from (87) and (88) is an explicit interacting representation of the Onsager algebra.

## 6.2 Conjectures on hidden Onsager algebra symmetries in spin-1 ZF models at root of unity

Motivated by the spin-1/2 case in Sec. 4.2 and the example of spin-1 $U(1)$-invariant clock model in Sec. 6.1, we arrive at exactly the same conjectures: (62), (65) and (66) with operators acting on spin-1/2 physical Hilbert space replaced by the ones acting on spin-1 physical Hilbert space. We shall not repeat the same equations here.

In the spin-1 case, Conjecture I (62) is proven for $\ell_2 = 3$, while the identification between the canonical generators $\mathbf{A}_m^r$ and the generators $\mathbf{Q}_m^r$ (63), (66) and (65) are checked numerically, cf. Sec. 6.1. Conjectures I and II have been verified numerically for various cases when roots of unity satisfy $\ell_2 = 3,4$ and all permitted values of $\ell_1$ with system size $N$ up to 10. Similar to the spin-1/2 case, the numerical evidence is convincing that Conjectures I and II are true for arbitrary root of unity value of the anisotropy and the system size.

## 7 Conclusion and outlook

In this article we focus on the hidden Onsager algebra symmetry structure in spin-1/2 XXZ model and its spin-1 generalisation, ZF model, at root of unity value of the anisotropy. By constructing the semi-cyclic transfer matrices and the generating functions for conserved charges, we propose two conjectures for the hidden Onsager algebra symmetries in the aforementioned models, motivated by two examples of spin-1/2 XX model and spin-1 $U(1)$-invariant clock model (ZF model with $\eta = \mathrm{i}\pi/3$). It is straightforward to observe that one can obtain similar results for higher spin generalisations of XXZ model at root of unity through transfer matrix fusion procedure, exemplified in Sec. 5. Despite the credibility of the conjectures, it would be interesting to prove them using quantum integrability, by means of the methods in [30,51]. The conjectures also hint at the relation between the underlying quantum group structure of those models at root of unity and Onsager algebra symmetry. Future investigations in this direction would reveal possible connections between them.

For spin-1/2 XXZ model at root of unity, we have two sets of commuting charges $\mathbf{Z}_m$ and $\mathbf{Y}_n$, while they do not commute with operators in the other set. The non-commutability between $\mathbf{Z}_m$ and $\mathbf{Y}_n$ has consequences in the thermodynamic limit, leading to oscillatory behaviour of auto-correlation functions [52,53]. The relation between the oscillatory behaviour of correlation functions in the thermodynamic limit to the hidden Onsager algebra symmetries in those models still needs further consideration.

Moreover, there are recent works on Onsager algebra and its q-deformation in XXZ model with open and half-infinite boundary conditions [54, 55]. It would be of great interest to understand the relation to the results in this article concerning the same model with quasi-periodic boundary condition. There are several generalisations of the Onsager algebra that have appeared in the literature [56–58], which are related to spin chains associated with higher-rank symmetries. It would be of great interest to investigate the physical applications for the generalisations of the Onsager algebra.

## Acknowledgements

I am very grateful to Jules Lamers and Vincent Pasquier for the previous collaborations on a related topic and numerous discussions. I thank Oleksandr Gamayun, Enej Ilievski, Jiří Minář and Eric Vernier for valuable discussions and critical remarks on the manuscript. I thank Paul Fendley, Hosho Katsura, Marko Medenjak and Lenart Zadnik for valuable discussions. I am grateful to the referees for the comments to improve the manuscript. I acknowledge the support from the European Research Council under ERC Advanced grant 743032 DYNAMINT.

## A  Isomorphism between $O$ and $O'$

We start with the standard presentation of the Onsager algebra $O$. We construct the following linear combinations of the generators $\mathbf{A}_m$ and $\mathbf{G}_n$ satisfying $O$,

$$\tilde{\mathbf{A}}_m^0 = \frac{1}{2}\left(\mathbf{A}_m + \mathbf{A}_{-m}\right), \quad \tilde{\mathbf{A}}_m^{\pm} = \frac{1}{4}\left(\mathbf{A}_m - \mathbf{A}_{-m}\right) \pm \frac{1}{2}G_m, \tag{A.1}$$

with $m \in \mathbb{Z}$. It is straightforward to check that new generators $\tilde{\mathbf{A}}_m^r$ satisfy precisely the presentation $O'$ by checking relations (3), (4) and (5). It implies that $O'$ is a subalgebra of $O$.

We switch to the presentation $O'$. We construct again the following linear combinations of the generators $\mathbf{A}_m^r$ satisfying $O'$,

$$\tilde{\mathbf{A}}_m = \mathbf{A}_m^0 + \mathbf{A}_m^+ + \mathbf{A}_m^-, \quad \tilde{\mathbf{G}}_m = \mathbf{A}_m^- - \mathbf{A}_m^+, \tag{A.2}$$

with $m \in \mathbb{Z}$. Those new generators satisfy the definition of $O$ (1). Thus it implies that $O$ is a subalgebra of $O'$.

Combining these two conclusions, we observe that $O$ and $O'$ are isomorphic to each other. They are both equivalent presentations of the Onsager algebra. We refer them to the presentation $O$ and the presentation $O'$ of the Onsager algebra, respectively.

## B  The Onsager algebra as a subalgebra of $\mathfrak{sl}_2$ loop algebra

The Onsager algebra is also a subalgebra of the $\mathfrak{sl}_2$ loop algebra $L(\mathfrak{sl}_2)$ [56]. This can be demonstrated in a straightforward manner using the presentation $O'$. Before showing that $O'$ (as well as $O$ due to the isomorphism) is a subalgebra of $L(\mathfrak{sl}_2)$, we start with the definition of $L(\mathfrak{sl}_2) \cong \mathfrak{sl}_2 \otimes \mathbb{C}[t, t^{-1}]$ with generators $\{\mathbf{e}_m^+, \mathbf{e}_m^-, \mathbf{f}_m | m \in \mathbb{Z}\}$ in Chevalley basis. $\mathbb{C}[t, t^{-1}]$ stands for the algebra consisting of all the Laurent polynomials with coefficients in the field of complex numbers $\mathbb{C}$. The $\mathfrak{sl}_2$ generators satisfy

$$[\mathbf{e}^+, \mathbf{e}^-] = \mathbf{f}, \quad [\mathbf{f}, \mathbf{e}^{\pm}] = \pm 2\mathbf{e}^{\pm}. \tag{B.1}$$

The generators of $L(\mathfrak{sl}_2)$ are thus

$$\mathsf{e}_m^\pm = \mathsf{e}^\pm \otimes t^m, \quad \mathsf{f}_m = \mathsf{f} \otimes t^m, \tag{B.2}$$

with $m \in \mathbb{Z}$. They satisfy the following relation,

$$[\mathsf{e}_m^+, \mathsf{e}_n^-] = \mathsf{f}_{n+m}, \quad [\mathsf{f}_m, \mathsf{e}_n^\pm] = \pm 2\mathsf{e}_{n+m}^\pm. \tag{B.3}$$

Using the generators of $L(\mathfrak{sl}_2)$, we construct the following generators

$$\tilde{\mathbf{A}}_m^0 = \mathsf{f}_m + \mathsf{f}_{-m}, \quad \tilde{\mathbf{A}}_m^\pm = \pm\left(\mathsf{e}_m^\pm - \mathsf{e}_{-m}^\pm\right). \tag{B.4}$$

Generators defined in (B.4) satisfy the definition of the presentation $O'$, cf. (3), (4) and (5), using (B.3). Therefore, the presentation $O'$ and its isomorphism $O$ are subalgebras of $\mathfrak{sl}_2$ loop algebra.

**Remark.** Spin-1/2 XXZ model at root of unity has been shown to possess the $\mathfrak{sl}_2$ loop algebra symmetries in [59, 60]. It might seem trivial to conjecture that spin-1/2 XXZ model at root of unity possesses the Onsager algebra symmetry, since the Onsager algebra is a subalgebra of the $\mathfrak{sl}_2$ loop algebra. However, it is not the case. The reasons are as follows. The $\mathfrak{sl}_2$ loop algebra generators for XXZ model at root of unity are defined differently for each magnetisation sectors $S^z = m(\mathrm{mod}\,\ell_2)$. For different values of $m$, the $\mathfrak{sl}_2$ loop algebra generators are different [59]. For instance, the simplest example is when magnetisation satisfying $S^z = 0(\mathrm{mod}\,\ell_2)$ with $q_0 = \exp\left(i\pi\frac{\ell_1}{\ell_2}\right)$. It is shown in Ref. [59] that operators

$$\left(\mathbf{S}^\pm\right)^{(\ell_2)} = \lim_{q\to q_0}\frac{1}{[\ell_2]_q}\left(\mathbf{S}^\pm\right)^{\ell_2}, \quad \left(\bar{\mathbf{S}}^\pm\right)^{(\ell_2)} = \lim_{q\to q_0}\frac{1}{[\ell_2]_q}\left(\bar{\mathbf{S}}^\pm\right)^{\ell_2} \tag{B.5}$$

are related to a representation of the algebra $L(\mathfrak{sl}_2)$ by making the following identification

$$\begin{aligned}
\mathsf{e}_0^\pm &= \left(\mathbf{S}^\pm\right)^{(\ell_2)}, \quad \mathsf{e}_1^\pm = \left(\bar{\mathbf{S}}^\pm\right)^{(\ell_2)}, \\
\mathsf{f}_0 &= -\mathsf{f}_1 = -\left(-q_0\right)^{\ell_2}\frac{\mathbf{S}^z}{\ell_2}.
\end{aligned} \tag{B.6}$$

The definitions of operators $\mathbf{S}^\pm$, $\bar{\mathbf{S}}^\pm$ and $q$-number can be found in Appendix C. These generators only commute with the XXZ Hamiltonian and transfer matrices within the sector $S^z = 0(\mathrm{mod}\,\ell_2)$. For other sectors $S^z \neq 0(\mathrm{mod}\,\ell_2)$, operators (B.6) are no longer the generators and there exist other operators in terms of $\mathbf{S}^\pm$ and $\bar{\mathbf{S}}^\pm$ that commute with the XXZ Hamiltonian and transfer matrices $\mathbf{T}_s$ within the specific sector only, which forms a representation of the algebra $L(\mathfrak{sl}_2)$. In short, the $\mathfrak{sl}_2$ loop algebra symmetries proposed in Ref. [59, 60] depend on the magnetisation sectors.

However, the conjectures presented in this article apply to all the states within the physical Hilbert space of $N$ sites. In another word, the generators for the Onsager algebra symmetries are the same for any magnetisation sectors. This is the crucial difference between the conjectures in this article and the preceding results in the literature. Already from here, we observe that the Onsager algebra symmetries conjectured are of close relation to the $\mathfrak{sl}_2$ loop algebra symmetries for different magnetisation sectors discussed previously. We postpone this discussions to investigations in the future.

# C   Representations of $\mathcal{U}_q(\mathfrak{sl}_2)$

In this appendix we briefly mention a few representations used in this article. For a mathematical and complete treatment of the representations of the Hopf algebra $\mathcal{U}_q(\mathfrak{sl}_2)$, we refer the readers to [32].

## C.1 Global representation

The physical Hilbert space $(\mathbb{C}^2)^{\otimes N}$ can be used to construct two global representations of $\mathcal{U}_q(\mathfrak{sl}_2)$, cf. (15). The coproduct can be defined in two ways,

$$\mathbf{S}^{\pm} \mapsto \mathbf{S}^{\pm} \otimes \mathbf{K}^{-1} + \mathbf{K} \otimes \mathbf{S}^{\pm}, \quad \mathbf{K} \mapsto \mathbf{K} \otimes \mathbf{K}, \tag{C.1}$$

or

$$\bar{\mathbf{S}}^{\pm} \mapsto \bar{\mathbf{S}}^{\pm} \otimes \mathbf{K} + \mathbf{K}^{-1} \otimes \bar{\mathbf{S}}^{\pm}, \quad \bar{\mathbf{K}} = \mathbf{K} \mapsto \mathbf{K} \otimes \mathbf{K}. \tag{C.2}$$

Counits and antipodes are defined accordingly, see e.g. Eqs. (1.2)–(1.4) in Ref. [23]. When the Hilbert space is $\mathbb{C}^2$, $\mathbf{S}^{\pm} = \bar{\mathbf{S}}^{\pm}$.

Explicitly, when the physical Hilbert space is $\mathbf{C}^2$, the representation is given by $\mathbf{S}^{\pm} = \sigma^{\pm}$ and $\mathbf{K} = q^{\sigma^z/2}$. For the physical Hilbert space $(\mathbb{C}^2)^{\otimes N}$ with $N > 2$ we obtain two (reducible) representations acting the two coproducts above

$$\mathbf{S}^{\pm} = \sum_{j=1}^{N} q^{\sigma_1^z/2} \otimes \cdots \otimes q^{\sigma_{j-1}^z/2} \otimes \sigma_j^{\pm} \otimes q^{-\sigma_{j+1}^z/2} \otimes \cdots \otimes q^{-\sigma_N^z/2},$$
$$\mathbf{K} = q^{\mathbf{S}^z} = q^{\sigma_1^z/2} \otimes q^{\sigma_2^z/2} \otimes \cdots \otimes q^{\sigma_N^z/2}, \tag{C.3}$$

and

$$\bar{\mathbf{S}}^{\pm} = \sum_{j=1}^{N} q^{-\sigma_1^z/2} \otimes \cdots \otimes q^{-\sigma_{j-1}^z/2} \otimes \sigma_j^{\pm} \otimes q^{\sigma_{j+1}^z/2} \otimes \cdots \otimes q^{\sigma_N^z/2},$$
$$\bar{\mathbf{K}} = q^{\bar{\mathbf{S}}^z} = q^{\sigma_1^z/2} \otimes q^{\sigma_2^z/2} \otimes \cdots \otimes q^{\sigma_N^z/2} = \mathbf{K}. \tag{C.4}$$

## C.2 Unitary representations of $\mathcal{U}_q(\mathfrak{sl}_2)$

Similarly to the $\mathfrak{sl}_2$ algebra, with any deformation parameter $q$, there exist $(2s+1)$-dimensional unitary representation of $\mathcal{U}_q(\mathfrak{sl}_2)$ algebra when spin satisfies $2s \in \mathbf{Z}_{\geq 0}$.

We can express the representations in the $(2s + 1)$-dimensional vector space $V_a$ spanned over $\{|n\rangle\}_{n=0}^{2s}$ with $2s \in \mathbf{Z}_{\geq 0}$. Specifically,

$$\mathbf{S}_a^z = \sum_{n=0}^{2s} (-s+n)|n\rangle\langle n|, \quad \mathbf{K}_a = \exp\left(\eta \mathbf{S}_a^z\right) = \sum_{n=0}^{2s} q^{-s+n}|n\rangle\langle n|, \tag{C.1}$$

$$\mathbf{S}_a^+ = \sum_{n=0}^{2s-1} \sqrt{[2s-n][n+1]}|n+1\rangle\langle n|, \quad \mathbf{S}_a^- = \sum_{n=0}^{2s-1} \sqrt{[2s-n][n+1]}|n\rangle\langle n+1|, \tag{C.2}$$

with $q$-number $[x] = (q^x - q^{-x})/(q - q^{-1})$. These representations are called "unitary" due to the fact that $\left(\mathbf{S}_a^+\right)^{\dagger} = \mathbf{S}_a^-$.

It is easy to verify that the $(2s+1)$-dimensional unitary representations satisfy the relation (15) by direct calculation.

## C.3 $\ell_2$-dimensional semi-cyclic representation of $\mathcal{U}_q(\mathfrak{sl}_2)$

When we consider the algebra $\mathcal{U}_q(\mathfrak{sl}_2)$ at root of unity $q = \exp\left(i\pi\frac{\ell_1}{\ell_2}\right)$, there always exists a $\ell_2$-dimensional semi-cyclic representation that satisfies the relation (15) [30,33].

The $\ell_2$-dimensional semi-cyclic representation, parametrised by complex spin $s \in \mathbb{C}$ and semi-cyclic parameter $\beta \in \mathbb{C}$, is defined on a $\ell_2$-dimensional vector space $V_a$ spanned over $\{|n\rangle\}_{n=0}^{\ell_2-1}$. Explicitly we have,

$$\mathbf{S}_a^z = \sum_{n=0}^{\ell_2-1}(-s+n)|n\rangle\langle n|, \quad \mathbf{K}_a = \exp\left(\eta\mathbf{S}_a^z\right) = \sum_{n=0}^{\ell_2-1} q^{-s+n}|n\rangle\langle n|, \tag{C.1}$$

$$\mathbf{S}_a^+ = \beta|0\rangle\langle\ell_2-1| + \sum_{n=0}^{\ell_2-2}[2s-n]|n+1\rangle\langle n|, \quad \mathbf{S}_a^- = \sum_{n=0}^{\ell_2-2}[n+1]|n\rangle\langle n+1|. \tag{C.2}$$

When parameter $\beta = 0$, the representation is no longer semi-cyclic, and it is called the $\ell_2$-dimensional highest-weight representation [30]. In that case, the $\mathcal{U}_q(\mathfrak{sl}_2)$ relation (15) is still satisfied.

# D    Choice of $u_0$ in (29)

In (29) we expand the generating functions $\mathbf{Z}(u, \phi)$ and $\mathbf{Y}(u, \phi)$ at different spectral parameter values for cases with $\varepsilon = \pm 1$. We would like to provide some details in this appendix. As usual, the twist $\phi$ satisfies commensurate condition (21).

To begin with, we notice that when $\varepsilon = +1$,

$$\mathbf{T}_{(\ell_2-1)/2}^{\text{sc}}\left(\frac{\eta}{2}, 0, \phi\right) \tag{D.1}$$

is not invertible (i.e. not full-ranked), while it is invertible when $\varepsilon = -1$. Meanwhile, transfer matrices with $\varepsilon = +1$ are related to transfer matrices with $\varepsilon = -1$. We can see that from the existence of a unitary gauge transformation $\mathbf{U}$ [30, 61]

$$\mathbf{U} = \exp\left(\mathrm{i}\pi\sum_{j=1}^{N}\frac{j}{2}\sigma_j^z\right), \tag{D.2}$$

such that

$$\mathbf{U}\mathbf{H}(\Delta, \phi)\mathbf{U}^\dagger = -\mathbf{H}(-\Delta, \phi'). \tag{D.3}$$

The twists are related as

$$\phi' = \begin{cases} \phi & N \text{ even,} \\ \phi + \pi & N \text{ odd.} \end{cases} \tag{D.4}$$

We consider 2 transfer matrices, one with $\eta$ ($\varepsilon = \exp(\ell_2\eta) = -1$) and the other one with $\eta' = \mathrm{i}\pi - \eta$ ($\varepsilon' = \exp(\ell_2\eta') = +1$). This implies that $\ell_2$ is odd. When $s = \frac{\ell_2-1}{2}$ and $\beta = 0$, we have

$$\mathbf{U}\mathbf{T}_s^{\text{sc}}(u, 0, \phi, \eta)\mathbf{U}^\dagger = \mathbf{T}_s^{\text{sc}}(-u, 0, \phi, \eta'). \tag{D.5}$$

The above equation is satisfied only when $s \in \mathbb{Z}_{>0}$. This implies that $\frac{\ell_2-1}{2} \in \mathbb{Z}_{>0}$, i.e. $\ell_2$ is odd. Moreover, if $\phi$ with parameter $\eta$ in (D.4) satisfies commensurate condition (21), $\phi'$ with parameter $\eta'$ in (D.4) also satisfies commensurate condition (21).

Therefore, if we were to define

$$\mathbf{Q}_1^0(\eta, \phi) = \mathbf{Z}_0 = \frac{1}{2\eta}\,\partial_s\log\mathbf{T}_s^{\text{sc}}(u, \beta, \phi, \eta)\big|_{s=(\ell_2-1)/2, \beta=0, u=\eta/2}, \tag{D.6}$$

it is natural to define

$$\begin{aligned}
\mathbf{Q}_1^0(\eta',\phi') &= \mathbf{U}\mathbf{Q}_0(\eta,\phi)\mathbf{U}^\dagger \\
&= \frac{1}{2\eta'}\,\partial_s \log \mathbf{T}_s^{\text{sc}}(u,\beta,\phi',\eta')\big|_{s=(\ell_2-1)/2,\beta=0,u=-\eta/2}\,,
\end{aligned} \tag{D.7}$$

satisfying the same algebraic relations after applying the unitary gauge transformation $\mathbf{U}$. Similar relations for $\mathbf{Q}_1^\pm$ can be inferred.

In this case $-\frac{\eta}{2} = \frac{\eta'}{2} - \frac{i\pi}{2}$, indicating (29). We have used the value of $u_0$ defined in (29) to numerically verify the conjectures in Sec. 4.2. For instance, for the cases of $\eta = 2i\pi/3$, $2i\pi/5$ and $4i\pi/5$, the conjectures remain true with system size $N$ up to 12.

# E  Onsager generators in XX case

In the case of XX model, we obtain analytically all the Onsager generators when the twist is commensurate, cf. (21) by calculating the recursion relation analytically. The results are as follows.

$$\mathbf{Q}_m^0 = \frac{i}{2}\sum_{j=1}^N (-i)^{m-1}\sigma_j^+ \sigma_{j+1}^z \cdots \sigma_{j+m-1}^z \sigma_{j+m}^- - i^{m-1}\sigma_j^- \sigma_{j+1}^z \cdots \sigma_{j+m-1}^z \sigma_{j+m}^+, \tag{E.1}$$

$$\mathbf{Q}_m^- = \frac{i}{2}\sum_{j=1}^N (-i)^{m-1}(-1)^j \sigma_j^- \sigma_{j+1}^z \cdots \sigma_{j+m-1}^z \sigma_{j+m}^- = \left(\mathbf{Q}_m^+\right)^\dagger, \tag{E.2}$$

where $\sigma_{N+k}^\pm = e^{\pm i\phi/2}\sigma_k^\pm$ with $1 \le k < N$. All generators are bilinear in fermionic operators after Jordan–Wigner transformation [16].

From the formulae (E.1) and (E.2), we observe that

$$\mathbf{Q}_{m+2N}^0 = \mathbf{Q}_m^0, \quad \mathbf{Q}_{m+2N}^\pm = \mathbf{Q}_m^\pm, \tag{E.3}$$

i.e. the closure condition in (52).

# F  Sketch of proof for the validity of boost operator approach

Boost operator approach can be used to obtain the (quasi-)local density of higher-order charges with arbitrary auxiliary space [41]. In the derivation below, we do not assume any constraint on the auxiliary space $a$. In practice, we consider the case when the densities of conserved charges under consideration are local. For the semi-cyclic transfer matrix, it requires that $\ell_2 = 2s + 1$ when the physical spin is $s$. We demonstrate the validity considering $s = 1/2$ ($\ell_2 = 2$), i.e. XX model. One can generalise the construction to higher-spin scenarios.

We start with the "RLL" relation for the semi-cyclic Lax operator,

$$\mathbf{L}_{am}^{\text{sc}}(u,s,\varepsilon\beta)\mathbf{L}_{an}^{\text{sc}}(u+v,s,\varepsilon^2\beta)\mathbf{R}_{mn}(v) = \mathbf{R}_{mn}(v)\mathbf{L}_{an}^{\text{sc}}(u+v,s,\varepsilon\beta)\mathbf{L}_{am}^{\text{sc}}(u,s,\varepsilon^2\beta), \tag{F.1}$$

where the R matrix reads

$$\mathbf{R}_{mn}(v) = \mathbf{L}_{mn}^{\text{sc}}(v+\eta/2, 1/2, 0). \tag{F.2}$$

The R matrix satisfies the following properties

$$\mathbf{R}_{mn}(0) = \sinh\eta\,\mathbf{P}_{mn}, \quad \mathbf{R}_{mn}^{-1}(0)\,\partial_v \mathbf{R}_{mn}(v)\big|_{v=0} = \frac{1}{\sinh\eta}\mathbf{h}_{m,n}, \tag{F.3}$$

where the local terms for the Hamiltonian (33) are

$$\mathbf{h}_{j,j+1} := \mathbf{h}_j, \quad \mathbf{H} = \sum_j \mathbf{h}_j. \tag{F.4}$$

We differentiate (F.1) with respect to spectral parameter $v$ and take the limit $v \to 0$, yielding

$$
\begin{aligned}
&\mathbf{L}_{am}^{sc}(u,s,\varepsilon\beta)\partial_u\mathbf{L}_{an}^{sc}(u,s,\varepsilon^2\beta) - \partial_u\mathbf{L}_{am}^{sc}(u,s,\varepsilon\beta)\mathbf{L}_{an}^{sc}(u,s,\varepsilon^2\beta) \\
&= -\mathrm{i}\left[\mathbf{h}_{m,n}, \mathbf{L}_{am}^{sc}(u,s,\varepsilon\beta)\mathbf{L}_{an}^{sc}(u,s,\varepsilon^2\beta)\right],
\end{aligned}
\tag{F.5}
$$

where

$$\lim_{v\to 0}\partial_v\mathbf{L}_{am}^{sc}(u+v,s,\varepsilon^2\beta) = \partial_u\mathbf{L}_{am}^{sc}(u,s,\varepsilon^2\beta). \tag{F.6}$$

Taking the following limit, $m \to j$, $n \to j+1$, and $\beta \to \varepsilon^{j-1}\beta$, we obtain

$$
\begin{aligned}
&\mathbf{L}_{aj}^{sc}(u,s,\varepsilon^j\beta)\partial_u\mathbf{L}_{a(j+1)}^{sc}(u,s,\varepsilon^{j+1}\beta) - \partial_u\mathbf{L}_{aj}^{sc}(u,s,\varepsilon^j\beta)\mathbf{L}_{a(j+1)}^{sc}(u,s,\varepsilon^{j+1}\beta) \\
&= -\mathrm{i}\left[\mathbf{h}_{m,n}, \mathbf{L}_{aj}^{sc}(u,s,\varepsilon^j\beta)\mathbf{L}_{a(j+1)}^{sc}(u,s,\varepsilon^{j+1}\beta)\right].
\end{aligned}
\tag{F.7}
$$

In the following, we sketch the essential steps to obtain the boost operator. First, we multiply on both sides of (F.7)

$$\prod_{k=1}^{j-1}\mathbf{L}_{ak}^{sc}(u,s,\varepsilon^k\beta), \tag{F.8}$$

from the left and

$$\prod_{k=j+1}^{N}\mathbf{L}_{ak}^{sc}(u,s,\varepsilon^k\beta), \tag{F.9}$$

from the right. Next, we take the trace over the auxiliary space $a$. Finally, we multiply by $(j+1/2)$ on both sides and sum over $j$. By telescoping the sum on the left hand side, we obtain

$$\partial_u\mathbf{T}_s^{sc}(u,\beta,\phi) = \mathrm{i}\left[\mathbf{T}_s^{sc}(u,\beta,\phi),\mathcal{B}\right], \tag{F.10}$$

where the boost operator is of form

$$\mathcal{B} = \sum_j \mathcal{B}_j, \quad \mathbf{B}_j = \frac{1}{\sinh\eta}\left(j+\frac{1}{2}\right)\mathbf{h}_j. \tag{F.11}$$

Here we also take the limit $N \to \infty$ to avoid the boundary terms.

From (F.10) we have

$$\partial_u\mathbf{Z}(u) = \mathrm{i}\left[\mathbf{Z}(u),\mathcal{B}\right], \tag{F.12}$$

$$\partial_u\mathbf{Y}(u) = \mathrm{i}\left[\mathbf{Y}(u),\mathcal{B}\right]. \tag{F.13}$$

Matching the coefficient for each order of $u^m$ using (28), we have

$$\mathbf{Z}_m = \frac{\mathrm{i}}{m}\left[\mathbf{Z}_{m-1},\mathcal{B}\right], \quad m \in \mathbf{Z}_{>0}, \tag{F.14}$$

$$\mathbf{Y}_m = \frac{\mathrm{i}}{m}\left[\mathbf{Y}_{m-1},\mathcal{B}\right], \quad m \in \mathbf{Z}_{>0}. \tag{F.15}$$

Using the boost operator (F.11), we check that the local density for higher-order charges is correct even in the presence of a diagonal twist with a finite system size. This procedure can be generalised to higher-spin cases in the same manner.

# G Proof of Lemma (56)

Starting from (E.1), we calculate the commutator with the boost operator (54) for $m \geq 2$,

$$
\begin{aligned}
\mathrm{i}\big[\mathbf{Q}_m^0, \mathcal{B}^{\mathrm{XX}}\big] &= \sum_j j\left(\frac{(-\mathrm{i})^{m-1}}{2}\sigma_{j-m}^+\sigma_{j-m+1}^z\cdots\sigma_j^z\sigma_{j+1}^- - \sigma_j^+\sigma_{j+1}^z\cdots\sigma_{j+m}^z\sigma_{j+m+1}^- + \mathrm{h.c.}\right) \\
&\quad + j\left(\frac{(-\mathrm{i})^{m-1}}{2}\sigma_{j-m}^+\sigma_{j-m+1}^z\cdots\sigma_{j-1}^z\sigma_j^- - \sigma_{j+1}^+\sigma_{j+2}^z\cdots\sigma_{j+m}^z\sigma_{j+m+1}^- + \mathrm{h.c.}\right) \\
&= m\sum_j\left(\frac{(-\mathrm{i})^{m-1}}{2}\sigma_j^+\sigma_{j+1}^z\cdots\sigma_{j+m}^z\sigma_{j+m+1}^- + \mathrm{h.c.}\right) \\
&\quad + m\sum_j\left(\frac{(-\mathrm{i})^{m-1}}{2}\sigma_j^+\sigma_{j+1}^z\cdots\sigma_{j+m-2}^z\sigma_{j+m-1}^- + \mathrm{h.c.}\right) \\
&= m\mathbf{Q}_{m+1}^0 - m\mathbf{Q}_{m-1}^0,
\end{aligned}
\tag{G.1}
$$

where we have telescoped the series after the second equals sign.

Similarly, from (E.2), we have

$$
\begin{aligned}
\mathrm{i}\big[\mathbf{Q}_m^-, \mathcal{B}^{\mathrm{XX}}\big] &= m\sum_j\left(\frac{(-\mathrm{i})^{m-1}}{2}(-1)^j\sigma_j^-\sigma_{j+1}^z\cdots\sigma_{j+m}^z\sigma_{j+m+1}^-\right) \\
&\quad + m\sum_j\left(\frac{(-\mathrm{i})^{m-1}}{2}(-1)^j\sigma_j^-\sigma_{j+1}^z\cdots\sigma_{j+m-2}^z\sigma_{j+m-1}^-\right) \\
&= m\mathbf{Q}_{m+1}^- - m\mathbf{Q}_{m-1}^-,
\end{aligned}
\tag{G.2}
$$

for $m \geq 2$.

When $m = 1$, we obtain

$$
\mathrm{i}\big[\mathbf{Q}_1^0, \mathcal{B}^{\mathrm{XX}}\big] = \frac{1}{2}\sum_j\left(\sigma_j^+\sigma_{j+1}^z\sigma_{j+2}^- + \sigma_j^-\sigma_{j+1}^z\sigma_{j+2}^+ - \sigma_j^z\right) = \mathbf{Q}_2^0 - \mathbf{Q}_0^0,
\tag{G.3}
$$

$$
\mathrm{i}\big[\mathbf{Q}_1^-, \mathcal{B}^{\mathrm{XX}}\big] = \frac{1}{2}\sum_j(-1)^j\sigma_j^-\sigma_{j+1}^z\sigma_{j+2}^- = \mathbf{Q}_2^- - \mathbf{Q}_0^-.
\tag{G.4}
$$

Therefore, we conclude that for all $m \in \mathbf{Z}_{>0}$,

$$
\mathrm{i}\big[\mathbf{Q}_m^0, \mathcal{B}^{\mathrm{XX}}\big] = m\mathbf{Q}_{m+1}^0 - m\mathbf{Q}_{m-1}^0,
\tag{G.5}
$$

$$
\mathrm{i}\big[\mathbf{Q}_m^-, \mathcal{B}^{\mathrm{XX}}\big] = m\mathbf{Q}_{m+1}^- - m\mathbf{Q}_{m-1}^-.
\tag{G.6}
$$

Taking the complex conjugate on (G.6), we obtain

$$
-\mathrm{i}\big[(\mathcal{B}^{\mathrm{XX}})^\dagger, \mathbf{Q}_m^+\big] = \mathrm{i}\big[\mathbf{Q}_m^+, \mathcal{B}^{\mathrm{XX}}\big] = m\mathbf{Q}_{m+1}^+ - m\mathbf{Q}_{m-1}^+,
\tag{G.7}
$$

which concludes the proof of Lemma (56).

# H  Higher-order Z and Y charges in spin-1 case

We present the result for the charges $Z_1$ and $Y_1$ in local density terms explicitly for spin-1 ZF model with $\eta = i\pi/3$,

$$
\begin{aligned}
Z_1 &= i\big[Z_0, \mathcal{B}^{\mathrm{ZF}}\big] \\
&= \frac{2}{3}\sum_j \Big[ S_j^z \mathcal{S}_j^- \mathcal{S}_{j+1}^+ + \mathcal{S}_j^+ S_j^z \mathcal{S}_{j+1}^- + \mathcal{S}_j^+ \mathcal{S}_{j+1}^- S_{j+1}^z + \mathcal{S}_j^- S_{j+1}^z - 2S_j^z \\
&\quad + \mathcal{S}_j^- S_{j+1}^z \mathcal{S}_{j+2}^+ + \mathcal{S}_j^+ S_{j+1}^z \mathcal{S}_{j+2}^- + \left(\mathcal{S}_j^-\right)^2 S_{j+1}^z \left(\mathcal{S}_{j+2}^+\right)^2 + \left(\mathcal{S}_j^+\right)^2 S_{j+1}^z \left(\mathcal{S}_{j+2}^-\right)^2 \\
&\quad - \mathcal{S}_j^- \{\mathcal{S}_{j+1}^-, S_{j+1}^z\} \left(\mathcal{S}_{j+2}^+\right)^2 - \mathcal{S}_j^+ \{\mathcal{S}_{j+1}^+, S_{j+1}^z\} \left(\mathcal{S}_{j+2}^-\right)^2 \\
&\quad - \left(\mathcal{S}_j^-\right)^2 \{\mathcal{S}_{j+1}^-, S_{j+1}^z\} \mathcal{S}_{j+2}^+ - \left(\mathcal{S}_j^+\right)^2 \{\mathcal{S}_{j+1}^+, S_{j+1}^z\} \mathcal{S}_{j+2}^- \Big] \\
&= \frac{3}{2}\left(Q_2^0 - Q_0^0\right);
\end{aligned}
\tag{H.1}
$$

$$
\begin{aligned}
Y_1 &= i\big[Y_0, \mathcal{B}^{\mathrm{ZF}}\big] \\
&= \frac{2}{3}\sum_j \Big[ \frac{1}{2}\left(\mathcal{S}_j^-\right)^2 \mathcal{S}_{j+1}^- S_{j+1}^z + \frac{1}{2}\left(\mathcal{S}_j^-\right)^2 S_{j+1}^z \mathcal{S}_{j+1}^- + \frac{1}{2}\mathcal{S}_j^- S_j^z \left(\mathcal{S}_{j+1}^-\right)^2 \\
&\quad + \frac{1}{2} S_j^z \mathcal{S}_j^- \left(\mathcal{S}_{j+1}^-\right)^2 + \left(\mathcal{S}_j^-\right)^2 S_{j+1}^z \mathcal{S}_{j+2}^- + \mathcal{S}_j^- S_{j+1}^z \left(\mathcal{S}_{j+2}^-\right)^2 \\
&\quad - \left(\mathcal{S}_j^-\right)^2 \{\mathcal{S}_{j+1}^+, S_{j+1}^z\} \left(\mathcal{S}_{j+2}^-\right)^2 - \mathcal{S}_j^- \{\mathcal{S}_{j+1}^-, S_{j+1}^z\} \mathcal{S}_{j+2}^- \Big] \\
&= \frac{3}{2}\left(Q_2^- - Q_0^-\right).
\end{aligned}
\tag{H.2}
$$

These two charges $Z_1$ and $Y_1$ are obtained using boost operator approach.

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
