# Peer review of "Conjectures on Hidden Onsager Algebra Symmetries in Interacting Quantum Lattice Models"

_SciPost Physics, doi:SciPost Phys. 11, 066 (2021)_

## Round 2 · Referee Report · Anonymous (Referee 1) · 2021-5-16

Strengths

  1. New non-trivial algebraic structure conjectured for the very well studied system

Weaknesses

Lack of proofs for the cases where the statement can be proved (free fermion point for the spin 1/2 chain, $\pi/3$ point for the spin 1 case)

Report

The main result of the paper is a conjecture for the hidden Onsager symmetry for spin 1/2 and spin 1 XXZ chains at roots of unity. The Onsager algebra generators are expressed in terms of the quasi-local charges. The result is new and can be extremely useful.
The conjecture is based on the observation for the free-fermion point for the spin $\frac 12$ case and $\pi/3$ point for the spin 1 case. As this is the main source of the conjectured general result the lack of detailed proof of the 2 conjectures for these simple points is the main reason why the paper cannot be published in its present form. The author also mentions that only a part of the conjecture 2 is proved for the free fermion case and which one is not, it should be stated in a more precise way.
As a conclusion : the paper should be certainly published once comprehensive derivations are added for the clock models (at least as appendices but better in the main part of the paper).

Requested changes

  1. Add proofs and detailed construction of the quasi-local charges in sections 4.1 and 6.1
  2. Give a detailed explanation which part of the Conjecture 2 is not yet proved for the simple cases.
  3. Add derivations for the equations (4.7), (4.9), (6.7) and (6.8).
  4. It would be useful to provide a proof of the closer condition (4.8) in the free fermion point (at least as an appendix)

---

## Round 2 · Referee Report · Anonymous (Referee 2) · 2021-5-22

Report

A. Overview and general comment

Two quantum integrable models are considered: the quasi-periodic (twist $\phi$) spin-1/2 XXZ chain, and its spin-1 generalization - the Zamolodchikov-Fateev model. For root of unity values of the deformation parameter $q$, it is conjectured that the Hamiltonian enjoys a hidden non-Abelian symmetry associated with (a quotient of, strictly speaking) the Onsager algebra. The conjectures are proposed based on the construction of transfer matrices for a semi-cyclic auxiliary space representation following [29].

This article contains promising results, extending further Vernier-O'Brien-Fendley recent results [15]. In particular, it suggests to investigate above models from the point of view of the Onsager algebra. This might open the possibility of investigating correlation functions and out-of-equilibrium dynamics in these models, see e.g. [16]. For these reasons, it deserves to be published in SciPost Physics after some revisions. The revisions mainly aim to improve the presentation and clarify few points. Details are given below.

Nota bene: Strictly speaking, the paper deals with different quotients of the Onsager algebra. For what is concerned in the paper, this point is not considered as essential. However, it may become for a deeper analysis of the model considered (see e.g. reference [R0] below).

B. Suggestions

  1. In the literature, the Onsager algebra $O$ admits different types of presentations through generators and relations. The historical one [1] is given in terms of generators $A_m,G_n$ with relations (2.1). Another presentation by Dolan-Grady [8] is given in terms of generators $A_0,A_1$ with (2.3). Now, in [15] a new algebra with relations (2.6)-(2.9) is introduced, with generators $A_m^{0},A_m^{\pm}$. To avoid any confusion, let's call this algebra $O'$. Following [15], the author gives an embedding $O \subset O'$ in (2.5). Thus, $O$ may be - from this information only - a subalgebra of $O'$, not necessarely equivalent to $O$. However, as written in the text of the present form, it is not clear for the reader how $O'$ and $O$ are related: are they isomorphic? is one a non-trivial subalgebra of the other? Clarifying that is not just a mathematical question: it is essential, as the key results in further sections are various representations for which $[H,O']=0$. Indeed, for instance without further clarification it may happen that there is no map $O' \rightarrow O$ ($O$ may be smaller than $O'$). In that case, the real symmetry (characterizing the fine structure of the model) would be $O'$, not $O$. Then, by (2.5) $[H,O]=0$ would not be essential. But that would contradict the claim that $O$ is the most interesting symmetry of the model. To clarify this issue, one needs to check that there exists an inverse map $O' \rightarrow O$. Looking at [15], this map can be constructed from the relations below (1.2) of [15]: each generator of $O'$ can be written solely in terms of $A_0,A_1$ of $O$.
    According to previous comments, $O'$ and $O$ (provided (2.5) holds) are indeed equivalent. So, as this point is crucial although no comments are given in the present form of the paper, I suggest the author to improve the sentence

From (2.1), it is easy to observe...

above (2.13) as follows (for instance):

`From [15], it is known that all generators $A_m^{0},A_m^{\pm}$ of (2.6)-(2.9) can be written as polynomials in $A_0,A_1$. Thus, from

$$[H,A_m^r]=0 \ , \ r \in {0,+,-}, \ m \in {\mathbb Z}\ ,$$

and (2.5) one finds (2.12).'

  1. Below (3.5), the notation ${\bf L}^{\textsf{sc}}_{aj}(u,s,\beta)$ is introduced. Please define $s$ (later on it is mentionned below (3.9), but that should be done below (3.5)). Also, to be self-contained and because ${\bf L}^{\textsf{sc}}_{aj}(u,s,\beta)$ plays a crucial role in the following analysis, the definition of ${\bf L}^{\textsf{sc}}_{aj}(u,s,\beta)$ should be clearly stated/improved. Also, it is written The transfer matrix is therefore denoted as ${\bf L}^{\textsf{sc}}$. I think it should beThe Lax operator is therefore denoted as $L^{\textsf{sc}}_{aj}(u,s,\beta)$'.

  2. Below (3.5), it is said that semi-cyclic representations of $U_q(sl_2)$ are condidered. Please add a precise reference (ref with eqs. number) where they are described explicitly. What are the expressions of ${\bf K}_a,{\bf S}^\pm_a$ in this case? Please add it somewhere in the text.

  3. Above (3.6), it is written 'As proven in [29],...'. However, in [29] I see (3.3) but can't find a proof of the claim. So the sentence should be modified. If it is not proven in the paper, a reference for the proof should be given.

  4. The proof that (3.7) solves (3.6) is not given. It is expected to be a corollary of (3.6), but for a non-expert reader, a reference is welcome.

  5. As a corollary of (3.6), it is expected that (3.8) are mutually commuting for arbitrary values of $u$. For a non-expert reader, a sentence and a reference are welcome.

  6. In (3.13), ${\bf T}_s(u,\phi)$ is introduced without definition. How is it related with ${\bf T}^{sc}_s(u,\beta,\phi)$? ${\bf T}_s(u,\phi)$ should be clearly defined below (3.8).

  7. Above (3.13)-(3.14), it is claimed that both relations hold. No assumptions on the parameter $\phi$ are specified. However, top of page 7 it is written 'Note that ..$Y$ charges satisfy (3.13)-(3.14) only when the twist $\phi$ is commensurate'. That is confusing. If (3.13)-(3.14) holds only for $\phi$ commensurate, this sentence top of page 7 should be right above (3.13).

  8. In the literature (physics and maths), the terminology Onsager generators' is standard, and always refers to $A_m,G_n$. In the paper, the author sometimes used the termOnsager generators' for $A_m,G_n$, but also for the new generators $A_m^{0},A_m^{\pm}$. Below (2.9), I suggest either to add a sentence explaining that in the text, the terminology Onsager generators' is also used for $A_m^{0},A_m^{\pm}$ (there may be still some readers for whom that will remain anyway confusing), or to introduce the terminologyOnsager type generators' for $A_m^{0},A_m^{\pm}$.

C. Additional references suggested, typos, cosmetic changes

  1. Introduction: "Later Onsager has been used" $\rightarrow$ "Later the Onsager algebra has been used".

  2. In the article, the Onsager algebra is generated from studying transfer matrices associated with RLL quantum Yang-Baxter algebras. In the literature, it has been shown recently that the Onsager algebra (and generalizations of [50]) arises from classical non-standard Yang-Baxter algebras [R1]. For completeness, it may be helpful to complete the sentence in the Introduction: "A thorough and comprehensive summary...[14]" by:

"Furthermore, recently an isomorphism between the Onsager algebra and a non-standard classical Yang-Baxter algebra is obtained [R1]".

  1. In the text, please replace when appropriate "Onsager algebra" $\rightarrow$ "the Onsager algebra" (ex: beginning of section 2); Below (2.2): "the Dolan-Grady (DG) relation" $\rightarrow$ "the Dolan-Grady (DG) relations" (indeed, one has two relations).

  2. In Figure 1: to be consistent with previous notations: "${\bf T}^{sc}$" $\rightarrow$"${\bf T}_a^{sc}(u,\beta,\phi)$".

  3. First line of Section 4. It is said "It is well-known that XX model ...possesses Onsager algebra symmetry." Please add a reference.

  4. In Section 7, it is written "Despite the credibility of the conjectures, it would be interesting to prove them using quantum integrability". Actually, it may happen that part of the analysis in the author's paper (and of [29]) share some similarity with the analysis and proofs in a series of papers of Shi-shyr Roan between 2006 and 2012. For instance, see reference [R2] below. If relevant, a comment may be added and adding few references would make sense.

  5. About the last sentence of Section 7. Actually, generalizations of Onsager algebra were first introduced by Uglov-Ivanov (A-type) in [R3], and Date-Usami (D-type) [R4]. I would recommend to add [R3,R4] together with [50]. It makes sense, not only for historical purpose. Indeed, SciPost is a physics journal, so connections between generalized Onsager algebras and integrable models - as pointed out in [R3] - would be helpful to the reader.

  6. Typo in Ref. [34]: "...$ofU(qsl2)in$..." $\rightarrow$ "of ...$U_q(sl_2)$... in".

D. References

[R0] B. Davies, Onsager's algebra and superintegrabilityDOI:10.1088/0305-4470/23/12/010Corpus ID: 119898494

[R1] Baseilhac, P., Belliard, S. Cramp\'e, N. FRT presentation of the Onsager algebras. Lett Math Phys 108, 2189–2212 (2018). https://doi.org/10.1007/s11005-018-1068-x

[R2] S-s Roan, The Transfer Matrix of Superintegrable Chiral Potts Model as the Q-operator of Root-of-unity XXZ Chain with Cyclic Representation of $U_q(sl_2)$, J.Stat.Mech.0709:P09021,2007. doi 10.1088/1742-5468/2007/09/P09021

[R3] D.B. Uglov, I.T. Ivanov, $sl(N)$ Onsager’s algebra and integrability, J. Stat. Phys. 82 (1996), 87–113.

[R4] E. Date, K. Usami, On an analog of the Onsager algebra of type $D_n^{(1)}$. In: ”Kac-Moody Lie algebras and related topics”, 43–51, Contemp. Math., 343, Amer. Math. Soc., Providence, RI, 2004.

Attachment

---

## Round 2 · Referee Report · Anonymous (Referee 3) · 2021-6-8

Strengths

1- Exhibits a remarkable feature (Onsager algebra symmetry) previously unknown in well-studied quantum integrable systems (XXZ chains at root of unity) 2- Self-contained and clearly written

Weaknesses

1- No explicit expression for the Onsager generators (except at the free fermion point), therefore not so clear whether the Onsager symmetry will be of much use for the present models 2- The presence of Onsager generators is almost everywhere conjectured, no solid proof.

Report

In this work the author builds upon a previous work of his and collaborators [29] as well as on [15] by another group, to conjecture on the presence of a hidden Onsager algebra symmetry in XXZ quantum chain at root of unity, as well as in its spin 1 descendent (the Fateev-Zamolodchikov chain). The Onsager generators are related to transfer matrices built out of semi-cyclic representations of the quantum group $U_q(sl_2)$, and are argued to account for the multiple degeneracies observed in the spectrum of such models.

While I'm afraid the results presented here might have limited applicability (in contrast with the earlier work [15] where the Onsager generators enjoyed an explicit expression in terms of local densities, here they are only quasilocal and never written explicitly), it is nice enough to see a precise characterization of the relation between the Onsager algebra and XXZ chains so that this paper deserves publication in SciPost, once taken in account the minor requests below.

Requested changes

1- there are some typos, and some awkward formulations, so I suggest that the author goes over the draft one more time and corrects the grammar. two examples : p1 : "Hamltonian". p10 : "one might wonder the question"

2- in the introduction, the phrasing "higher spin generalizations of XX model" is a bit deceiptive, as it might suggest that the higher spin models are, as the XX model, free. Rather, the naming "higher spin generalizations the XXZ model at root of unity"

3- Section 2 is confusing in many respects, in my opinion. The first reason is the naming "Onsager algebra" for different algebras, as already pointed out by a previous referee. This should be clarified. The second reason is that this section is presented as a self-contained and generic exposition of the Onsager algebra, but it includes some statements which refer to applications to XXZ chains (to which the rest of the paper is devoted) without this being mentioned explicitly, hence causing confusion. In particular, the statement "$A_0$ and $A_1$ are considered as U(1) charges" is unclear, as well as the statement that Dolan-Grady relation imply Kramers-Wannier self-duality. It would be more fair to say, in my opinion, that in the following of the paper $A_0$ has an incarnation as a U(1) charge identified as the magnetization, and that the existence of $A_1$ and DG relations suggests the existence of some sort of duality.

4- a question for the author : for the spin 1 chain (p. 13), the author mentions the special case eta=i pi/3, where the charges can be written in terms of local densities. Might not it be the case also for eta=2ipi/3 ?

5- finally, could the author clarify somewhere, or at least have a word about, the relation between this Onsager algebra symmetry, and the loop group $L(sl_2)$ which is a known feature of root-of-unity chains ?

---

## Round 3 · Referee Report · Anonymous (Referee 2) · 2021-7-2

Strengths

Conjectured Onsager algebra symmetry for the quasi-periodic XXZ spin chain at roots of unity

Report

We thank the author for the corrections. The draft still needs to be improved, some important points remain to be clarified. Suggestions for changes are given below.

  1. In the begining of section 4.2, the author implicitly identifies the Q's with the A's. That is OK for the XX model studied in section 4.1, as shown by (4.7)-(4.10) which supports the identification (4.12) and the use of the terminology 'Onsager generators' for the Q's'. However, that is not clear for the XXZ model at root of unity. For the Q's from (4.29)-(4.30), one should check that the relations of O' in order to claim the Q's as 'Onsager generators'. So, Conjecture I should be improved. It has two parts. One is the observation (4.28) holds. That's OK, numerical evidences are supporting this, as explained below (4.33). But the second part of the conjecture is that (4.12) extends to the XXZ case. However, it is not clear if the author has checked that (4.29)-(4.30) satisfy the defining relations for O'. So, the following changes should be done:
  2. above (4.29), it should be said that (4.12) is conjectured.
  3. A reference to eq. (4.4) should be added above (4.31).
  4. Below conjecture I, a comment about the numerical check of (4.12) should be added.

  5. Similarly to point 1, in section 6.2 a comment on the numerical check of (4.12) should be added.

  6. Strictly speaking, in the Ising model the Onsager generators A0,A1 map to combinations of Pauli matrices according to (2.7). Clearly, the r.h.s of (2.7) generates a quotient of O: besides the defining relations for O, additional relations arise (shown by Davies years later). From that point of view, (2.7) should be improved by replacing equalities by --> (\rightarrow). Similarly, in the text the Q's are images of the Onsager generators in some quotient of the Onsager algebra. So, the Q's satisfy additionnal relations besides the ones for O or O'. In the text, it would be better to put '\rightarrow' instead of equalities when appropriate. For (4.12), it would be better to write

$A^r_m \rightarrow \frac{\ell_2}{4} Q^r_m$.

  1. This comment may not be seen as important if the reader is a physicist, but might be for a mathematician. And because O' is new, it may be relevant for future works on the subject. The Onsager algebra has various presentations. The standard presentation denoted O, and the new one denoted O' by the author. So in the text, instead of saying 'the Onsager algebra O' and the 'Onsager algebra O'', it is better to specify at the begining of section 2 that two presentations of the Onsager algebra are used: the standard one denoted O and the other denoted O'. Then, in the Appendix although O and O' are a priori distinct at the begining, the proof shows that they are actually two presentations of the same object. Then, it is better to conclude that O and O' are two presentations of the Onsager algebra.

  2. The wording should be double checked again. For instance: below (3.6), 'relaiton' --> 'relation'; below (4.19), 'Combing' --> 'Combining'.

  3. The typo in ref. [36] 'ofUq(sl2)in' --> 'of $U_q(sl_2)$ shlould be corrected.

  4. In [51], typo $u_q(sl_2)$ --> $U_q(sl_2)$

---

## Round 3 · Referee Report · Anonymous (Referee 1) · 2021-7-6

Report

The author has implemented changes required in my previous report. I recommend the paper to be published after correcting some typos.

Requested changes

  1. Correct a typo in the eq. 6.11

---

## Round 3 · Referee Report · Anonymous (Referee 3) · 2021-7-13

Report

I thank the author for having improved the manuscript, which I now consider ready for publication in SciPost.

---

## Round 3 · Author Response

I am grateful to the Referees for their assessment of our manuscript and for their valuable and detailed comments and suggestions. My replies to the queries and list of changes are given below.

  1. I have added the proof of the Onsager algebra symmetries for XX model and spin-1 $U(1)$ invariant clock model using boost operator approach. The explicit expressions and recursive formulae for the higher-order Z/Y charges are derived and presented in the new appendices.

  2. As the referee suggested, I show the isomorphism between the two algebras which are now called algebra $O$ and $O^\prime$ in the draft, see Appendix A. This provides the justification of denoting algebra $O^\prime$ as the Onsager algebra, which has been used already in Ref. [16] but not explicitly shown.

  3. As the referee suggested, I add the discussion about the relation between the Onsager algebra and the $\mathfrak{sl}_2$ loop algebra in Appendix B. Specifically, I clarify the difference between the previously proposed the $\mathfrak{sl}_2$ loop algebra symmetries of XXZ model at root of unity and the conjectured Onsager algebra symmetries of XXZ model at root of unity in this draft.

  4. As the referee pointed out, the spin-1 model with anisotropy parameter $\eta = \mathrm{i} \pi /3$ can be mapped into the same model with anisotropy parameter $\eta = 2 \mathrm{i} \pi /3$ via a unitary transformation. This is stated clearly in the first paragraph of Section 6 now.

  5. I add the definition of $\mathbf{T}_s (u, \phi)$ and I move the commensurate condition for the twist $\phi$ before defining the Z/Y charges. I also complement Appendix C with the representations of $\mathcal{U}_q (\mathfrak{sl}_2)$ that have been used in the draft.

  6. I change the typos and grammatical mistakes in the main text and bibliography. The references that have been suggested by the referees are added too, which I am grateful for the referees to point out this.

I would like to thank the referees again for the great suggestions. Should there be any more improvement, I would be appreciated to receive the comments.

---

## Round 3 · List of Changes

1. I have added the definition of the transfer matrix $\mathbf{T}_s$ in Sec. 2.

  2. I have added new proofs on the Onsager symmetries for XX model and spin-1 $U(1)$-invariant clock model using boost operator approaches. These are included in Sec. 4 and 6. A short review on the boost operator approach is added in App. F.

  3. I have added the proof of the isomorphism of the two Onsager algebras mentioned in the draft, $O$ and $O^\prime$ in App. A.

  4. A short discussion between Onsager algebra and $\mathfrak{sl}_2$ loop algebra is added in App. B.

  5. Additional references that are relevant to the draft are included.

  6. Several typos and grammatical errors are fixed.

---

## Round 4 · Referee Report · Anonymous · 2021-7-19

Report

The author has corrected all points rised in the previous report. I do recommend this new revised version for publication.

---

## Round 4 · Referee Report · Anonymous · 2021-7-20

Report

I recommend this version for publication in SciPost.

---

## Round 4 · Author Response

Dear referees and editor,

I am grateful to the Referees for for their valuable comments and suggestions. The list of changes are given below.

---

## Round 4 · List of Changes

1. I have changed the Eq. (4.12) as suggested by the referee. I added Eq. (4.33) in the new version to stress the conjectural nature of the identifications between the canonical generators and generators $\mathbf{Q}_m^r$. I have performed the numerical check that Eq. (4.29) and (4.30) indeed satisfy the presentation $O^\prime$ using the identification of Eq. (4.31). A reference to Eq. (4.4) has been added in front of Eq. (4.32) (in the previous version Eq. (4.31) ).

2. A similar comment about the identification between the canonical generators and generators $\mathbf{Q}_m^r$ in Section 6.2 has been added.

3. In Eq. (2.7) and (4.12), I have changed the expressions according to the referee's suggestion. I also added the comment about the representation of the quotient of the Onsager algebra in the transverse field Ising model in the line below Eq. (2.7).

4. I have changed the notations on the presentations of the Onsager algebra $O$ and $O^\prime$ according to the referee's suggestion throughout the draft.

5. I have changed several typos in the main text and the references, especially the one in Eq. (6.11).

---

## Editorial Decision

published